# Phytoplankton size class in the East China Sea derived from MODIS satellite data

Hailong Zhang[1,2], Shengqiang Wang[1,2], Zhongfeng Qiu[1,2], Deyong Sun[1,2], Joji Ishizaka[3], Shaojie Sun[4], Yijun He[1,2]

[1] School of Marine Sciences, Nanjing University of Information Science & Technology, Nanjing, Jiangsu, China

[2] Jiangsu Research Centre for Ocean Survey Technology, NUIST, Nanjing, Jiangsu, China

[3] Institute for Space-Earth Environmental Research, Nagoya University, Nagoya, Japan

[4] College of Marine Science, University of South Florida, St. Petersburg, Florida, USA

*Correspondence to*: Z. F. Qiu (zhongfeng.qiu@nuist.edu.cn)

**Abstract.** The distribution and variation of phytoplankton size class (PSC) are key to understanding ocean biogeochemical processes and ecosystem. Remote sensing of the PSC in the East China Sea (ECS) remains a challenge, although many algorithms have been developed to estimate PSC. Here based on a local dataset from the ECS, a regional model was tuned to estimate the PSC from the spectral features of normalized phytoplankton absorption ($a_{ph}$) using a principal component analysis approach. Before applying the refined PSC model to MODIS (Moderate Resolution Imaging Spectroradiometer) data, reconstructing satellite remote sensing reflectance ($R_{rs}$) at 412 and 443 nm was critical through modeling them from $R_{rs}$ between 469 and 555 nm using multiple regression analysis. Satellite-derived PSC results compared well with those derived from pigment composition, which demonstrated the potential of satellite ocean color data to estimate PSC distributions in the ECS from space. Application of the refined PSC model to the reconstructed MODIS data from 2003 to 2016 yielded the seasonal distributions of the PSC in the ECS, suggesting that the PSC distributions were heterogeneous in both temporal and spatial scales. Micro-phytoplankton were dominant in coastal waters throughout the year, especially in the Changjiang estuary. For the middle shelf region, the seasonal shifts from the dominance of micro- and nano-phytoplankton in the winter and spring to the dominance of nano- and pico-phytoplankton in the summer and autumn were observed. Pico-phytoplankton were especially dominant in the Kuroshio region in the spring, summer, and autumn. The seasonal variations of the PSC in the ECS were probably affected by a combination of the water column stability, upwelling, sea surface temperature, and the Kuroshio Current. Additionally, human activity and riverine discharge might also influence the PSC distribution in the ECS, especially in the coastal region.

# 1 Introduction

Phytoplankton size class (PSC) is fundamentally important for ocean biogeochemical processes and ecosystems, especially for photosynthesis efficiency (Bouman et al., 2005; Uitz et al., 2008), primary production, and the carbon transport (Kiørboe, 1993; Guidi et al., 2009; Hirawake et al., 2011). Thus, knowledge of the PSC dynamics can contribute to the improvement of

our understanding of marine ecological and biogeochemical cycles. The classical size fractions of phytoplankton proposed by Sieburth et al. (1978) include three classes, namely, micro- (> 20 μm), nano- (2-20 μm), and pico-phytoplankton (< 2 μm). Among the methods to measure PSC from water samples, including microscopy (Montagnes et al., 1994), Coulter counter method (Sheldon and Parsons, 1967), and flow cytometry (Sun et al., 2000), pigment concentrations by high-performance liquid chromatography (HPLC) is the most systematic and qualify-controlled method (Van Heukelem and Hooker, 2011).

However, these methods are time-consuming and methodologically complex. Furthermore, large spatial and temporal variabilities make it difficult to continuously monitor PSC using the field sampling methods.

Realistically, satellite ocean color data can provide synoptic observations, which are ideal for investigating PSC at large spatial and temporal scales. In recent years, various algorithms have been designed to estimate PSC using *in situ* data and ocean color data on both global and regional scales (IOCCG, 2014). Most algorithms can be partitioned into two categories, namely,

"abundance-based" and "spectral-based" methods. The "abundance-based" methods are based on the statistical relationship between phytoplankton size fraction and phytoplankton abundance using measurements such as chlorophyll-a concentration (Chla) (refer to Bracher et al. (2017) Table 2). These approaches rely on the assumption that high and low Chla waters are dominated by large and small phytoplankton, respectively. The "spectral-based" methods utilize the relationship between the variations in inherent optical properties with changes in the PSC using measurements such as phytoplankton absorption ($a_{ph}$),

remote sensing reflectance ($R_{rs}$), and particulate backscattering ($b_{bp}$) (refer to Bracher et al. (2017) Table 2).

The East China Sea (ECS) is the base of the marine fishery resources in China and is one of the most productive ocean areas in the world (Furuya et al., 1996). Ascertaining the distribution of PSC can provide valuable information on the state of marine ecosystem and primary production in the area. Recent efforts have been focused on investigating the phytoplankton community and size classes in the ECS and have suggested that the PSC exhibited obvious spatiotemporal heterogeneity in this region (Li

et al., 2007; Luan et al., 2007; Jiang et al., 2014). For instance, Chen (2000) investigated the PSC and primary productivity in the marginal regions of the southern ECS using field data. The results showed that the phytoplankton size structure and their contributions to primary production displayed significant spatial differences in the shelf waters, upwelling waters, and Kuroshio water. Furuya et al. (2003) presented the phytoplankton dynamics in the ECS in the spring of 1994 and the summer

of 1996 using HPLC-derived pigment signatures. A distinct horizontal heterogeneity in phytoplankton composition was observed in the spring, and a "two-layer" distribution of phytoplankton appeared both off and on the shelf in the summer. Liu et al. (2016) used 7-years (2006-2012) field measurements to investigate the seasonal and spatial variations of major phytoplankton groups in the ECS, and found that monsoon forcing was a key factor to impact phytoplankton dynamics at seasonal scale.

Note that previous investigations on the PSC in the ECS have been conducted based on field observations, which may not reflect the real variation patterns of PSC. To our knowledge, no study has attempted to examine the PSC distributions in the ECS at synoptic scales from satellite observations. Consequently, the PSC dynamics in the ECS at different spatial and temporal scales and their mechanisms are still poorly understood. In the ECS, Wang et al. (2014) found that the correlation between the variation patterns of the PSC and total Chla was not valid, and pointed out that the "abundance-based" methods for estimating PSC were probably not applicable in the ECS. Therefore, Wang et al. (2015) proposed a model to estimate the PSC in the ECS using the spectral shape of normalized $a_{ph}(\lambda)$ through principal component analysis (PCA). This model showed good performance for estimating the PSC from both *in situ* measured $a_{ph}$ and $R_{rs}$. However, this model was developed using field dataset mainly from offshore waters of the ECS and off the coastal Japan; more importantly, the Wang et al. (2015) model has not been implemented in satellite data yet.

Therefore, the goals of this study were to: (1) refine the Wang et al. (2015) model for regional application in the ECS using an extensive dataset covering highly varied water conditions and various seasons, (2) apply the refined PSC model to Moderate Resolution Imaging Spectroradiometer (MODIS) satellite data, and (3) then preliminarily investigate previously unknown seasonal and spatial variation patterns of the PSC in the ECS.

## 2 Materials and methods

### 2.1 Study area and sampling stations

The East China Sea is one of the largest marginal seas in the western North Pacific and is bounded by China, Korea, and Japan (Fig. 1a). Nearly 70% of the ECS is occupied by a continental shelf shallower than 200 m. Numerous rivers flow into the ECS from mainland China, including the Changjiang (Yangtze) River which provides nearly 90% of the riverine discharge to the ECS (Zhang et al., 2007). In addition, the ECS experiences strong currents and multiple water masses, such as Changjiang diluted water (CDW), shelf mixed water, and the Kuroshio Current (Ichikawa and Beardsley, 2002; Su and Yuan, 2005). Here we analyzed the mean shape and coefficient of variation (CV) of *in situ* $R_{rs}(\lambda)$ collected in the ECS, to better show the ocean color variability in the ECS which covers many water types (Fig. 2). The *in situ* $R_{rs}(\lambda)$ of all samples exhibited large variability

in both magnitudes and spectral shapes (Fig. 2a and b). For the samples in the coastal region of Zhejiang (Zhe) and Fujian (Min), both 10 wavebands showed larger variability in $R_{rs}(\lambda)$ magnitude with CV larger than 55% (Fig. 2c). For the samples in southern Jeju Island, CV varied from 20% to 60%, with a minimum around 531 nm and 547 nm (Fig. 2d). Overall, they showed large dynamic range with significant variability. Because of highly variable environmental conditions, the ECS exhibits complex marine biogeochemical processes and ecosystem.

The field measurements used in this study were collected from approximately 10 cruises over the last decade. These sampling stations were distributed irregularly in the ECS and a few were in the Tsushima Strait (Fig. 1a). The field dataset encompassed various seasons and environmental conditions of the ocean, including turbid waters in the mouth area of Changjiang river, less turbid coastal water, and clear water away from the coast. This field dataset consisted of *in situ* measured $a_{ph}(\lambda)$, measured $R_{rs}(\lambda)$ data, and phytoplankton pigments measured by HPLC. In total, 69 samples with synchronous measurements of pigments, $a_{ph}$, and $R_{rs}$ data, 101 samples with coincident pigments and measured $a_{ph}$ data, and 27 samples with only measured $R_{rs}$ were available, and Fig. 1a shows the spatial distribution of samples. The Kuroshio water in our study area suffered from a paucity of *in situ* $R_{rs}$ data. Hence, in addition to the regional dataset, 227 *in situ* $R_{rs}$ samples collected in the North Pacific and North Atlantic oceans (Fig. 1b) from the NASA SeaBASS archive were used as a supplementary dataset. The SeaBASS dataset was only used for algorithm development to reconstruct satellite $R_{rs}$ data, along with our regional field dataset (see Section 2.4). The average spectral shapes of the 14-years (2003-2016) MODIS $R_{rs}$ data in the North Pacific and North Atlantic oceans were similar to that in the Kuroshio water (Fig. 1c). Thus, *in situ* measured $R_{rs}$ data collected in the North Pacific and North Atlantic oceans were used in the present study, although the distribution regions of these data were beyond our study area.

Meanwhile, three specific subareas were selected for further investigation in this study, including the mouth area of Changjiang river (MCJR, 122.3-123.5 °E and 31-32 °N), middle shelf region (MSR, 123.5-125 °E and 28-29 °N), and Kuroshio region (KR, 126-127.1 °E and 25.2-26.2 °N), as marked by black boxes in Fig. 1a. These subareas were selected based on geographical locations and driving forces. Within each subarea, the averages of all valid values were calculated for further analysis.

**2.2 In situ measurements**

Surface water samples (0-3 m) were collected with Niskin samplers mounted on a CTD Rosette or a clean bucket. These water samples were used for measurements of $a_{ph}(\lambda)$ and pigment concentrations.

**2.2.1 Measurement and analysis of HPLC-derived PSC**

For pigment analysis, seawater samples were filtered onto 47 mm Whatman GF/F glass fiber filters under gentle pressure

(<0.01 MPa), and then stored initially on board in liquid nitrogen (-70 °C) for later analysis in the laboratory. Briefly, the concentrations of 19 pigments were determined by reverse-phase HPLC following Van Heukelem and Thomas (2001). To remove measurements with lower precision, the quality control (QA) process was applied to the pigment dataset by HPLC according to the rules of Aiken et al. (2009). In our study, the diagnostic pigment analysis (DPA) was applied to compute the

PSC values from HPLC pigment data (hereafter called the HPLC-derived PSC). In brief, the DPA approach uses seven diagnostic pigment concentrations to obtain the HPLC-derived PSC, including fucoxanthin ($C_f$), peridinin ($C_p$), 19′-hexanoyloxyfucoxanthin ($C_h$), 19′-butanoyloxyfucoxanthin ($C_b$), alloxanthin ($C_a$), Chlorophyll-b ($C_{Cb}$), and zeaxanthin ($C_z$). The DPA approach was originally proposed by Vidussi et al. (2001), and subsequently improved by Uitz et al. (2006). In addition, Hirata et al. (2008) used the improved DPA approach to account for the occurrence of $C_{Cb}$ in nano-phytoplankton

class, because $C_{Cb}$ was most abundant at high Chla (> 0.25 mg m$^{-3}$) and was a minor pigment at lower Chla. Subsequently, Brewin et al. (2010) and Hirata et al. (2011) further refined the DPA approach to account for ambiguity of $C_f$ signal in diatoms and the occurrence of $C_h$ signal in picophytoplankton. In this study, the HPLC-derived PSC were then given by:

$$f_{micro} = \left(1.41C_f + 1.41C_p\right)/\sum W_i P_i \tag{1}$$

$$f_{nano} = \left(0.60C_a + 0.35C_b + 1.01C_{Cb} + x \times 1.27C_h\right)/\sum W_i P_i \tag{2}$$

$$f_{pico} = \left(0.86C_z + y \times 1.27C_h\right)/\sum W_i P_i \tag{3}$$

where $f_{micro}$, $f_{nano}$, and $f_{pico}$ denote the size fractions of micro-, nano-, and pico-phytoplankton, respectively. $x$ and $y$ are the proportions of nano- and pico-phytoplankton in Hex, respectively. When Chla >0.08 mg m$^{-3}$, $x=1$ and $y=0$; when Chla are between 0.001 and 0.08 mg m$^{-3}$, $x=12.5$Chla and $y=1-12.5$Chla. $\sum W_i P_i$ is the weighted sum of the seven diagnostic pigments (Uitz et al., 2006), according to the formula:

$$\sum W_i P_i = 1.41C_f + 1.41C_p + 0.60C_a + 0.35C_b + 1.27C_h + 0.86C_z + 1.01C_{Cb} \tag{4}$$

### 2.2.2 Measurement of $a_{ph}$

To obtain $a_{ph}$ data, we used the quantitative filter technique (QFT) via a series of processes (Mitchell, 1990). Water samples were filtered through 25 mm Whatman GF/F glass fiber filters under gentle pressure, and immediately frozen on board in liquid nitrogen. In this study, the "transmittance" approach was used for the samples collected from southern Jeju Island and

the Tsushima Strait (hereafter referred to as dataset-1). The optical density (OD) values of total particles were measured using a dual-beam multi-purpose spectrophotometer between 350 nm and 750 nm at 1 nm resolution. Similarly, we measured the OD value of the detritus after extracting phytoplankton pigments in methanol at least 24 h. Meanwhile, a blank filter saturated

with pure seawater was used as the reference filter. Then, the absorption coefficients of total particles $a_p(\lambda)$ and detritus $a_d(\lambda)$ were calculated from the corresponding OD values based on a correction of Cleveland and Weidemann (1993). The "transmittance-reflectance" approach was performed on the samples collected from the coastal and offshore regions of Zhejiang, Fujian, and Jiangsu (hereafter referred to as dataset-2). The optical densities of the total particles, detritus, and reference filter were obtained in both "transmission mode" and "reflection mode" between 250-850 nm at 1 nm resolution using a PerkinElmer lamda650s. Then, we converted these OD values into $a_p(\lambda)$ and $a_d(\lambda)$ values using the method of Tassan and Ferrari (1995; 2002). Finally, the $a_{ph}$ data were obtained as the difference of $a_p(\lambda)$ - $a_d(\lambda)$ at all sampling stations.

### 2.2.3 Measurement of $R_{rs}$

To obtain $R_{rs}$ data in dataset-1, the PRR-800/811 was used to measure the vertical profiles of the downwelling irradiance $E_d(\lambda,z)$ and upwelling radiance $L_u(\lambda,z)$ at 13 spectral channels (380, 412, 443, 465, 490, 510, 532, 555, 565, 589, 625, 665, and 683 nm). The water-leaving radiance $L_w(\lambda)$ was then determined from the profile of $L_u(\lambda,z)$ (Hirawake et al., 2011). The above-water surface downwelling irradiance $E_d(\lambda,0^+)$ was simultaneously measured by a cosine collector. Then, $R_{rs}(\lambda)$ data were calculated as the ratio of $L_w(\lambda)$ to $E_d(\lambda,0^+)$. For the purpose of consistency with satellite observations that characterize the oceanic surface layer, our analysis exclusively considered the near-surface $R_{rs}$ data.

For dataset-2, $R_{rs}$ data was collected under suitable solar illumination (generally between 9am and 3pm local time) using an ASD FieldSpec spectroradiometer in the spectral range of 350-1050 nm with 1.5 nm increments. The radiance spectra of water, sky, and a gray reference panel were measured following the above-water measurement approach (Mueller et al., 2003). For each of the three targets, ten spectra were collected and then averaged after removing abnormal spectra. According to the Ocean Optics Protocol (Mueller et al., 2003), the $R_{rs}(\lambda)$ data were obtained as

$$R_{rs}(\lambda) = \left(L_t - \gamma * L_{sky}\right) / \left(L_p * \pi / \rho_p\right) \tag{5}$$

where $L_t$, $L_{sky}$, and $L_p$ correspond to the radiance values measured from the water, sky, and reference panel, respectively. $\rho_p$ is the diffuse reflectance of the reference panel provided by the manufacturer. $\gamma$ is the surface Fresnel reflectance related to wind speed (2.6% - 2.8% for 10 m s$^{-1}$ wind, 2.5% for <5 m s$^{-1}$ wind, 2.2% for calm weather) (Tang et al., 2004).

All $R_{rs}$ and $a_{ph}$ data were resampled at the centers of MODIS wavebands (i.e., 412, 443, 469, 488, 531, 547, 555, 645, 667, and 678 nm) using the spectral response function of the MODIS sensor.

### 2.3 Satellite data

The global standard monthly MODIS remote sensing reflectance, chlorophyll-a concentration, and sea surface temperature

(SST) products (Level 3, about 4 km resolution) from 2003 to 2016 were provided by the NASA Ocean Color website (http://oceancolor.gsfc.nasa.gov/). The dataset corresponding to our study area (25-35° N and 118-132° E) were extracted from these global coverage datasets. These regional $R_{rs}$ products were processed using the MathWorks MATLAB software to obtain the satellite-derived PSC. Additionally, daily Level 2 $R_{rs}$ data from MODIS sensor (1 km resolution) were downloaded from the NASA Ocean Color website.

Samples were matched to daily $R_{rs}$ data to assess the accuracy of satellite-derived $a_{ph}$ and PSC results. To ensure the validity of satellite data before the matchup analysis, the following constraints were applied to the matchup dataset: (1) the matchup dataset only included satellite data with an overpass time window within 5 h before and after the field measurements; (2) to reduce the effect of outliers, median $R_{rs}$ value for a window of size 3 centered on the sampling station coordinates was defined as satellite $R_{rs}$ data; (3) negative MODIS $R_{rs}$ data were eliminated from the matchup analysis. Based on these criteria, 21 satellite matchups with coincident measured $R_{rs}$, and 22 satellite matchups with coincident measured PSC and $a_{ph}$ were available, as shown in Fig. 1a.

**2.4 Model accuracy assessment**

To evaluate the consistency between the derived and measured values, the Pearson correlation coefficient ($R$), root mean square error (RMSE), and mean absolute percentage error (MAPE) were used. Statistical assessments were performed in $\log_{10}$ space for the phytoplankton absorption coefficient and in linear space for the phytoplankton size class. These statistical indicators can be written as:

$$RMSE = \frac{1}{n}\sqrt{\sum_{i=1}^{n}\left[\left(x_{i,\text{derived}} - x_{i,\text{field}}\right)/x_{i,\text{field}}\right]^2} \tag{6}$$

$$MAPE\left(\%\right) = \frac{1}{n}\sum_{i=1}^{n}\left|\left(x_{i,\text{derived}} - x_{i,\text{field}}\right)/x_{i,\text{field}}\right| \times 100\% \tag{7}$$

where $n$ is the number of samples. $x_{i,\text{derived}}$ and $x_{i,\text{field}}$ are the derived and measured data for the $i$-th sampling station, respectively.

**2.5 Modified the Wang et al. (2015) model for retrieving PSC**

Wang et al. (2015) developed an spectral-based PSC model to quantify the size fractions of three phytoplankton classes using the spectral shape of $a_{ph}(\lambda)$ through PCA approach. Details of the development and parameterization of the model were described in Wang et al. (2015). In brief, to reduce the biomass effects, the normalized $a_{ph}(\lambda)$ (hereafter called $a_{ph}^{std}(\lambda)$) was computed by the ratio of $a_{ph}(\lambda)$ to their wavelength mean values in the spectral range between 412 and 547 nm. Then, the PCA approach was applied to the $a_{ph}^{std}(\lambda)$ to capture the spectral variation in phytoplankton absorption related to cell size. The input

of PCA is a $m \times N$ matrix constituted of $a_{\mathrm{ph}}^{\mathrm{std}}(\lambda)$, where $m$ and $N$ are the number of input wavelengths and samples, respectively. The output of PCA comprises two terms, i.e., principal component (PC) scores and PC weights (also called loading factors). The PC scores were assumed to correlate with the size class. Therefore, the relationships between the size fractions of micro- and pico-phytoplankton and PC scores were established using a logistic-type regression model (Hosmer Jr et al., 2013), as follows:

$$f_t = 1 \bigg/ \left[1+\exp\left(-\beta_0 - \sum_{i=1}^{k}\beta_i S_i\right)\right], \quad S_i = \sum_{j=1}^{m} w_{ij} a_{ph}^{std}(\lambda_j) \tag{8}$$

where $f_t$ denotes the phytoplankton size fraction ($t$ = micro or pico). $\beta_0$ and $\beta_i$ are the regression coefficients between $f_t$ and PC scores. $k$ is the number of PC scores ($k$ = 4 in this study). $w_{ij}$ refers to the loading factor for the $i$-th PC. $m$ is the number of wavelengths. Similar to previous studies (Brewin et al., 2010; Hirata et al., 2011), $f_{\mathrm{nano}}$ was calculated as 1-$f_{\mathrm{micro}}$-$f_{\mathrm{pico}}$, by considering that the sum of three phytoplankton size fractions was 1.

The $a_{\mathrm{ph}}(\lambda)$ at MODIS wavelengths were derived from $R_{\mathrm{rs}}(\lambda)$ data using the quasi-analytical algorithm (QAA) proposed by Lee et al. (2002). QAA was used in this study because it does not suppose a fixed shape for $a_{\mathrm{ph}}(\lambda)$ (Lee et al., 2002; 2009). Because QAA could give satisfactory retrievals of $a_{\mathrm{ph}}(\lambda)$ at the first 6 MODIS wavebands (i.e., 412, 443, 469, 488, 531, and 547 nm), as shown later for details, only $a_{\mathrm{ph}}(\lambda)$ data at these wavebands were used for the PSC model development in this study (i.e., $m$ = 6 in Eq. (8)).

To improve the accuracy of MODIS $R_{\mathrm{rs}}(\lambda)$ at short wavelengths (see details in Section 3.3), the reconstruction approach (Lee et al., 2014; Sun et al., 2015) was used to reconstruct satellite $R_{\mathrm{rs}}(\lambda)$ at 412 and 443 nm before applying of the refined PSC model to satellite data. In our study, satellite $R_{\mathrm{rs}}(412)$ and $R_{\mathrm{rs}}(443)$ were quantified as multivariable linear relationship using $R_{\mathrm{rs}}$ data from 469 to 555 nm, as follows:

$$R_{rs}^{rc}(\lambda) = \sum_{i=1}^{n} K_i R_{rs}(\lambda_i) + K_0 \tag{9}$$

where $R_{\mathrm{rs}}^{\mathrm{rc}}(\lambda)$ is the reconstructed $R_{\mathrm{rs}}$ data at wavelength $\lambda$ (412 or 443 nm); $R_{\mathrm{rs}}(\lambda_i)$ are the input $R_{\mathrm{rs}}$ data at five MODIS wavebands ($\lambda_i$ = 469, 488, 531, 547, and 555 nm); $K_0$ and $K_i$ are the coefficients determined from multivariant regression.

## 3 Results

### 3.1 Regional tuning of the PSC model for the ECS

Following Wang et al. (2015), Eq. (8) was fitted to 170 pairs of the HPLC-derived PSC and *in situ* measured $a_{\mathrm{ph}}$ data using a non-linear least square fitting procedure for developing the PSC model. The established parameters and associated $R$ and

RMSE values for each of the fits are shown in Table 1. Fig. 3 shows the strong linear relationships between the *in situ* $a_{ph}^{std}$-derived PSC and HPLC-derived results, with $R$ values of 0.89, 0.70, and 0.84 and RMSE values of 0.11, 0.11, and 0.11 for micro-, nano-, and pico-phytoplankton, respectively. The samples were close to the 1:1 line, with most of the samples within the ± 20% fraction range.

Using 69 measurements of $R_{rs}$ and associated HPLC-derived PSC and *in situ* measured $a_{ph}$, we also examined the feasibility of the PSC model for satellite observations by coupling QAA. First, we used QAA version 5 (QAA_v5) to retrieve $a_{ph}$ from measured $R_{rs}$. To assess the performance of QAA_v5, negative retrieved $a_{ph}$ values were eliminated, and the remainder were compared with the measured values at all MODIS wavebands (Fig. 4). The retrieved $a_{ph}$ values by QAA_v5 show reasonably good agreement with the *in situ* measured $a_{ph}$ at short wavelengths from 412 to 547 nm, with high $R$ values and low RMSE

and MAPE values. In contrast, the performance of QAA_v5 was poor and produced large overestimation of $a_{ph}$ at long wavelengths, especially at 645, 667, and 678 nm, consistent with previous findings (Lee et al., 2014; Tiwari and Shanmugam, 2014). These results clearly demonstrated that QAA_v5 can produce accurate estimates of $a_{ph}$ at 412, 443, 469, 488, 531, and 547 nm. Therefore, only $a_{ph}$ values at these bands were used to calibrate the PSC model, as previously stated. Then, the PSC values were inferred from the retrieved $a_{ph}$ using Eq. (8) with the established parameterizations. As shown in Fig. 5, the QAA

$a_{ph}^{std}(\lambda)$-derived PSC values were consistent with the HPLC-derived results, and almost all of the points fell within the ± 20% fraction range. The $R$ and RMSE values were 0.79 and 0.13 for micro-phytoplankton, 0.43 and 0.12 for nano-phytoplankton, and 0.80 and 0.13 for pico-phytoplankton, respectively. These results suggested that the refined PSC model for the ECS coupling QAA_v5 is able to accurately estimate the PSC from remote sensing reflectance $R_{rs}$.

**3.2 Comparison of satellite $R_{rs}$ with *in situ* measurements**

Before applying the PSC model to MODIS data, we assessed the accuracy of MODIS $R_{rs}$ using the synchronous *in situ* measurements. Table 2 shows the statistical results of the comparison between satellite $R_{rs}$ and *in situ* values for MODIS wavebands. For MODIS $R_{rs}$ data, a reasonably good consistency was found at green and red bands (from 469 to 555 nm), with $R$ values within 0.85 - 0.97 and MAPE values within 14.9% - 27.25%. Although the $R$ values were above 0.85, the MAPE values were high (> 54%) at 645, 667, and 678 nm. This is probably caused by the lower $R_{rs}$ values at these bands due to strong

absorption of water itself. In addition, a low accuracy was observed at 412 and 443 nm. The $R$ values were 0.46 and 0.73, and the MAPE values were 47.33% and 36.90% at 412 and 443 nm, respectively. The high noise and low accuracy at these two wavebands were suggested to be caused by the uncertainty of the atmospheric correction procedures and significant band degradation (Meister, 2011; Hu et al., 2013). Considering the importance of $R_{rs}(412)$ and $R_{rs}(443)$ to QAA algorithm, the poor

accuracy of satellite $R_{rs}$ at these bands may introduce uncertainty in the retrieved $a_{ph}$ data, and further increase the uncertainty of satellite-derived PSC. Thus, an accurate assessment of satellite-derived PSC requires the improved quality of satellite $R_{rs}$ data. In this study, the reconstruction approach was used to fulfill this objective.

### 3.3 Reconstruction of MODIS $R_{rs}$ data

The reconstruction function (Eq. (9)) was applied to the regional field dataset and SeaBass dataset to obtain the regression coefficients. The resulting relationships between the *in situ* measured and modeled $R_{rs}$ show strong agreement, with high $R^2$ and low RMSE and MAPE values (Table 3). For 412 and 443 nm, the $R^2$ values were close to 1.0 with a significance level of $p < 0.001$. The MAPE values were both lower than 9.0%. The reconstruction functions with the established coefficients were applied to the original MODIS $R_{rs}$ data to obtain the reconstructed satellite $R_{rs}^{rc}(412)$ and $R_{rs}^{rc}(443)$ data. Table 4 shows the comparison of the original satellite $R_{rs}$ and satellite $R_{rs}^{rc}$ data with *in situ* measured $R_{rs}$ at 412 and 443 nm. The satellite $R_{rs}^{rc}$ data were in better agreement with the *in situ* measured $R_{rs}$ data than the original satellite $R_{rs}$, especially at 412 nm. At 412 nm, the values of $R$, RMSE, and MAPE reached 0.70, 0.0019, and 35.15% for the satellite $R_{rs}^{rc}$ data, respectively, while these values were 0.46, 0.0026, and 47.33% for the original satellite data, respectively. These results indicated that the accuracy of the satellite $R_{rs}$ data at 412 and 443 nm could be improved through reconstruction using the selected MODIS wavebands.

### 3.4 Validation of satellite-derived $a_{ph}$ and PSC with *in situ* measured data

Based on the above analysis, we used the satellite $R_{rs}^{rc}(412)$ and $R_{rs}^{rc}(443)$ data rather than original satellite $R_{rs}$ data to compute $a_{ph}$ using QAA_v5. Fig. 6 shows the comparison of the derived $a_{ph}$ data from satellite $R_{rs}^{rc}$ (hereafter called $a_{ph}^{rc}$) and the derived $a_{ph}$ from original satellite $R_{rs}$ with *in situ* measurements at the first six MODIS wavebands. Table 5 summarized their corresponding statistical comparisons, i.e., $R$, RMSE, MAPE, and percentage of valid points (PVP). Here PVP is defined as the ratio of the number of positive satellite-derived values ($n$) to the total number of matchups ($N$) (as PVP = $n/N \times 100\%$). For the satellite-derived $a_{ph}^{rc}$, the $R$ values were above 0.80, except at 547 nm ($R = 0.69$), and were significantly higher than those for the satellite-derived $a_{ph}$ (with most of the values below 0.7). The statistics (RMSE, MAPE, and PVP) for the satellite-derived $a_{ph}^{rc}$ were also generally better than those for the satellite-derived $a_{ph}$. Compared with the satellite-derived $a_{ph}$, the PVP for the satellite-derived $a_{ph}^{rc}$ significantly increased with an average of 23.48%. Meanwhile, Fig. 6 also shows that the satellite-derived $a_{ph}^{rc}$ had more valid samples and were more clustered around the 1:1 line than the satellite-derived $a_{ph}$. Overall, both Table 5 and Fig. 6 indicated that the satellite-derived $a_{ph}$ had poor accuracy and low PVP values, whereas the accuracy of satellite-derived $a_{ph}^{rc}$ can be significantly improved with more valid samples through the reconstruction of satellite $R_{rs}$ data.

The refined PSC model was applied to the satellite-derived $a_{ph}$ data from original satellite $R_{rs}$ to estimate PSC (Fig. 7a). It can be seen that the satellite-derived PSC from original satellite $R_{rs}$ were inconsistent with the HPLC-derived results, showing obvious under and overestimations of the retrieved PSC for most of the samples. Their $R$ values were all below 0.27 (Fig. 7a). For comparison, we also estimated the PSC from the satellite-derived $a_{ph}^{rc}$ from reconstructed $R_{rs}^{rc}$ data and were compared with the HPLC-derived values (Fig. 7b). The satellite-derived PSC from reconstructed $R_{rs}^{rc}$ data agreed well with the HPLC-derived results. Their $R$ values of 0.68, 0.46, and 0.64 and RMSE values of 0.13, 0.13, and 0.19 were observed for micro-, nano-, and pico-phytoplankton, respectively. Almost all of the samples fell within the ±20% fraction range, although a slight underestimation of pico-phytoplankton size fraction occurred in a few samples.

Additionally, to further examine the performance of the refined PSC model in our study, our refined PSC model was compared with other two published PSC models (the Brewin et al. (2015) model and the Sun et al. (2017) model) (Fig. 7c and d). Here, we regionally tuned these published models using a standard nonlinear least-squares method based on our field dataset collected in the ECS. It should be noted here that the two retuned models were used to better assess the performance of our refined PSC model only, although these "abundance-based" models may not perform well in the ECS (data not shown) as suggested by Wang et al. (2014). In this study, the retuned Brewin et al. (2015) model for the ECS was expressed as:

$$
\begin{aligned}
f_{pico} &= 0.19\left[1 - exp\left(-3.6Chla\right)\right]/Chla \\
f_{p,n} &= 1.0\left[1 - exp\left(-1.0Chla\right)\right]/Chla \\
f_{nano} &= f_{p,n} - f_{pico} \quad and \quad f_{micro} = 1 - f_{p,n}
\end{aligned}
\tag{10}
$$

where $f_{p,n}$ is the sum of nano- and pico-phytoplankton size fraction. And, the retuned Sun et al. (2017) model for the ECS was expressed as:

$$
\begin{aligned}
f_{pico} &= 0.66Chla^{-1}\left[1 - exp\left(-Chla^2 \times R_{rs}\left(680\right)\right)\right]^{0.16} \\
f_{nano} &= 4.17Chla^{-1}\left[1 - exp\left(-Chla^2 \times R_{rs}\left(680\right)\right)\right]^{0.32} \\
f_{micro} &= 1 - f_{nano} - f_{pico}
\end{aligned}
\tag{11}
$$

where $R_{rs}$ (680) is the remote sensing reflectance at 680 nm. The scatter distributions of the satellite-derived PSC using our refined PSC model were closer to the 1:1 line than those of the other two models. According to the statistical indicators, our refined PSC model had the best performance, with higher $R$ values and lower RMSE values (Fig. 7b). For the retuned Brewin et al. (2015) model, the $R$ values of 0.58, 0.066, and 0.53 and RMSE values of 0.2, 0.14, and 0.18 were observed for micro-, nano-, and pico-phytoplankton, respectively (Fig. 7c). For the retuned Sun et al. (2017) model, the $R$ values were 0.36, -0.042, 0.5 for micro-, nano-, and pico-phytoplankton, when the corresponding RMSE values were 0.25, 0.17, and 0.18, respectively

(Fig. 7d). The retuned Brewin et al. (2015) model and the retuned Sun et al. (2017) model had relatively poor performance in the ECS. These comparison results indicated that the performance of our refined PSC model using the reconstructed satellite data was better than those of the retuned Brewin et al. (2015) model and the retuned Sun et al. (2017) model in our study region.

Overall, these results suggested that the use of satellite $R_{rs}^{rc}$ could significantly improve the performance of the refined PSC model on satellite observations and yielded reasonable satellite-derived PSC results, which were better than those derived from original satellite observations. Therefore, we further investigated the spatiotemporal variability of the PSC in the ECS based on satellite-derived products from the reconstructed satellite remote sensing reflectance.

**3.5 Seasonal distribution patterns of the PSC in the ECS**

To describe the seasonal variability of the PSC in the ECS, the refined PSC model was applied to 14 years (2003-2016) of MODIS monthly $R_{rs}$ data to obtain monthly PSC products. Then, seasonal composite PSC images were generated by averaging the monthly PSC products over a three month period for each season (Fig. 8 *a*-l). In this study, spring, summer, autumn, and winter were defined as March to May, June to August, September to November, and December to February of the next year, respectively. Meanwhile, to better understand the spatiotemporal variations of PSC, we analyzed the seasonal distributions of

Chla in the ECS for four season, as shown in Fig. 8 A-D.

Seasonal distributions of Chla (Fig. 8 A-D) illustrated that Chla were higher (0.4-3.0 mg m$^{-3}$) on the ECS shelf than in the Kuroshio water (<0.4 mg m$^{-3}$), and the Chla values in the Changjiang river mouth were particularly high (3.0-25 mg m$^{-3}$). During spring, the high Chla (>1.0 mg m$^{-3}$) were found on the ECS shelf, and the tongue-shape structure was unclear because of the increase of Chla in the surrounding areas. During summer, the Chla-values above 1.0 mg m$^{-3}$ were observed in the

coastal region. The higher Chla (>3.0 mg m$^{-3}$) were limited to the regions at the depth shallower than 30 m isobath, including the Changjiang mouth. In the autumn, the Chla remained high in the coastal region (>2.0 mg m$^{-3}$). The tongue-shaped structure extended outward the southeast along the 50 m isobath during autumn and along the 70 m isobath during winter.

Seasonal variation patterns of the PSC (Fig. 8 a-l) indicated that the phytoplankton size classes in the ECS were heterogeneous in both temporal and spatial scales. Their general distribution patterns were consistent with results reported from field

measurements by other researchers (Chen, 2000; Furuya et al., 2003; Wang et al., 2015). In the spring (Fig. 8 a-c), the higher $f_{micro}$ values (0.45 - 0.85) were found on the ECS shelf sea with lower values in offshore waters. Relatively high $f_{nano}$ (0.4 - 0.6) were clearly observed on offshore shelf and in southern Japan. However, pico-phytoplankton were the dominant size class over the southeastern ECS ($f_{pico}$ = 0.50 - 0.75). During summer (Fig. 8d-f), the micro-phytoplankton size fractions were still

high in coastal waters. The high $f_{micro}$ tongue-shape structure near the Changjiang Bank extended toward southeast along the 30 m isobath. High nano-phytoplankton proportions occurred in the ECS shelf sea with water depths of 30 - 200 m. The pico-phytoplankton contributions to Chla were relatively high around the ECS shelf break. Pico-phytoplankton represented the most abundant size class in the areas deeper than 200 m ($f_{pico} > 0.6$), which was similar to the results from the field measurements by Chen (2000). In the autumn (Fig. 8g-i), the $f_{micro}$ remained high in coastal waters and extended over the area shallower than 50 m isobath. The proportion patterns of nano- and pico-phytoplankton in the autumn were broadly similar to those in the summer. However, high nano-phytoplankton proportions were also in the northern Japan. In the winter (Fig. 8j-l), high $f_{micro}$ were mainly distributed on the ECS shelf. The regions with higher $f_{micro}$ (> 0.5) extended outward, and connected to the area around the Korean coast. The distributions of the size fractions of nano- and pico-phytoplankton were broadly similar to those in the spring.

## 3.6 Regional difference in the monthly climatological PSC in the ECS

Since the East China Sea is extensive, with a number of different environmental conditions and ecosystems, three subareas were selected for further investigation as shown in Fig. 1a. Within each of the subareas, this study investigated averages of the monthly climatological PSC, as well as chlorophyll-a concentration (Fig. 9).

In the MCJR, higher Chla were observed throughout the year (>3.0 mg m$^{-3}$), and two Chl-a peaks occurred in the spring (May) and summer (June) respectively (Fig. 9a). Throughout the year, micro-phytoplankton comprised 60% - 80% of the Chla, with the maximum value in April and relatively low fractions from summer to early autumn (June - September). Nano-phytoplankton comprised 18% - 30% of the Chla, while the contributions of pico-phytoplankton to Chla were below 10% throughout the year (Fig. 9a). In the MSR, mean Chla in this region domain were lower than those in the MCJR, with a peak in the spring (April) (Fig. 9b). The micro-phytoplankton proportions were slightly larger than pico-phytoplankton in the winter and spring, while the opposite was found in the summer and autumn. The pico-phytoplankton in the MSR was highest in August and September, with a peak in the summer and early autumn (June-September). Nano-phytoplankton were dominant (40% - 50%) for most of the year in this region (Fig. 9b). In contrast to the MCJR and MSR, the mean Chla were much lower in the Kuroshio region throughout the year (< 0.3 mg m$^{-3}$) (Fig. 9c). The KR domain showed a predominance of pico-phytoplankton (40% - 90%) throughout the year, with higher proportions observed in the summer. The nano-phytoplankton proportions (about 40%) were slightly lower than pico-phytoplankton in the winter and early spring, while their proportions became low (< 20%) in the rest of the year. The micro-phytoplankton size fractions in the KR remained low (< 23%) throughout the year (Fig. 9c).

## 4 Discussion

### 4.1 Satellite application of the refined PSC model

The most important advantage of satellite ocean color data is the ability to provide information on the spatiotemporal variability of the PSC. However, remote sensing of the PSC in the ECS is still a challenging task, although many "abundance-based" and "spectral-based" algorithms have been designed using field measurements and satellite data in the global scale. Taking into account the optical property in the ECS, Wang et al. (2014) reported that the "abundance-based" approaches are not necessarily applicable in the ECS, and the absorption spectra of phytoplankton could instead be used to obtain the PSC in the ECS. More than 80% of the variability in the spectral shape of phytoplankton absorption was highly related to the changes in the size classes (Ciotti et al., 2002; Bricaud et al., 2004). Therefore, in this study we refined the Wang et al. (2015) model for deriving PSC in the ECS from the spectral variation of $a_{ph}$. However, the application of this refined PSC model to original MODIS data has hampered, as showed in Fig. 7a. This may be related to the low accuracy of the MODIS $R_{rs}$ at 412 and 443 nm (Table 2), which can introduce additional uncertainties into the satellite-derived $a_{ph}$ from original $R_{rs}$ (Fig. 6; Table 5), and thereby affect the estimation accuracy of the satellite-derived PSC (Fig. 7a). To solve this problem, the multivariable linear relationship was employed to reconstruct MODIS $R_{rs}(412)$ and $R_{rs}(443)$ values using satellite $R_{rs}$ from 469 to 555 nm. Previous studies have reported that the use of multiple spectral bands could successfully reconstruct hyperspectral $R_{rs}$ data (Lee et al., 2014; Sun et al., 2015). In our study, the use of satellite $R_{rs}^{rc}$ improved the accuracy and PVP of the satellite-derived $a_{ph}^{rc}$ data using QAA_v5 (Fig. 6; Table 5), and dramatically improved the accuracy of the satellite-derived PSC (Fig. 7b). The $R$ and RMSE values for all size fractions derived from the reconstructed satellite $R_{rs}$ data were 0.7 and 0.15 respectively, compared to the values of 0.064 and 0.38 respectively for those derived from original satellite $R_{rs}$ data. Overall, this study successfully estimated the PSC in the ECS from the reconstructed MODIS remote sensing reflectance. The findings presented here complement recent studies that have demonstrated that satellite ocean color data can be used to retrieve the PSC in the ECS (Wang et al., 2015; Sun et al., 2017). However, it should be noted that there was no assessment of the credibility of satellite-derived PSC results in the Kuroshio waters due to lack of field dataset in this region, and further investigations focusing on the applicability of the reconstruction algorithm and the refined PSC model in Kuroshio waters and other regions are still required.

### 4.2 Spatial and temporal variations of the PSC in the ECS

As described in the results section 3.5 and 3.6, the seasonal distributions of the PSC and Chla in the East China Sea (Fig. 8 and Fig. 9) had great variability spatially and temporally. In general, micro-phytoplankton are favoured under environmental

condition with stronger mixing and high nutrient, while pico-phytoplankton are dominated in low-nutrient waters (IOCCG, 2014; Lamont et al., 2018). Here, we discussed the regional scale characterization of the full seasonal cycle in stellate-derived PSC and Chla and the related physical and biochemical effects for helping to understand the spatiotemporal variability of the PSC in the ECS.

### 4.2.1 The coastal region

In the coastal region, such as the coast of Zhejiang and Jiangsu (including the mouth area of Changjiang River), the combined effects of variable wind forcing, riverine discharge, and vertical mixing of the water column, promote phytoplankton growth, resulting in high biomass levels and the presence of larger-sized phytoplankton (Zhou et al., 2008; Wang et al., 2014). Seasonal distributions and variability of Chla (Fig. 8A-D and Fig. 9a) in the coastal region presented in this study generally agreed well
with the patterns reported by previous studies of satellite Chla (Yamaguchi et al., 2012; He et al., 2013).

In the spring, increased solar radiation and air temperature gradually warm up SST, which can reduce the vertical mixing of the water column. At the same time, weak wind stress can retain mixing of the water column, which transports nutrients to the upper layer from the nutrient-rich deep layer (Behrenfeld et al., 2006; Boyce et al., 2010). Meanwhile, coastal nutrient transporting to the inner shelf of the ECS can be enhanced under the north-westerly wind action (Liu and Wang, 2013). These
physical processes can allow phytoplankton to live longer in the upper euphotic layer in the sufficient nutrient and light conditions (Zhang et al., 2017), resulting in the spring bloom in the coastal region and inner shelf of the ECS (Fig. 8A), consistent with a previous study by Liu et al. (2016) based on the field measurement of Chla. This phenomenon was also clearly seen in Fig. 9a showing the monthly climatological Chla in the MCJR with a local maximum in April and May. These enhanced nutrient conditions favour the presence of micro-phytoplankton in the Changjing Bank and coastal waters (Fig. 8a
and Fig. 9a) and nano-phytoplankton in the offshore region (Fig. 8b). This was consistent with previous studies which showed that the large cell sizes, such as diatoms and *Prorocentrum donghaiense*, were dominant on the ECS shelf sea in the spring (Furuya et al., 2003; Lou and Hu, 2014; Liu et al., 2016).

In the summer, the coastal region with shallower than 40 m isobath displayed higher Chla (Fig. 8B) and higher micro-phytoplankton proportion (Fig. 8d). In the mouth area of Changjiang River, a long-lasting summer Chla maximum form May
to August was found. This may be related to the enhanced nutrient concentrations from river and estuarine discharges, e.g., the Changjiang River, Qiantangjiang River, and Minjiang River (Guo et al., 2014). Due to anthropogenic activities such as various agricultural and industry activities, nutrient-rich waters discharge into the East China Sea, especially in the summer monsoon rainy season (Siswanto et al., 2008). This is especially evident in higher Chla concentrations and higher micro-phytoplankton

proportions observed in the MCJR (Fig. 9a). Meanwhile, the littoral currents, e.g., the Zhe-Min Coastal Current (ZMCC) and Yellow Sea Coastal Current (YSCC), may play a key role in the transport of nutrient from riverine discharge. Previous studies reported that much of sediments is transported southward along the Zhejiang-Fujian coast by the ZMCC (Liu et al., 2007). In addition, the coastal region is relatively shallow, and the water body therefore has a weak stratification of water column in the summer. The hydrodynamic in coastal waters is dominated by the variation of wind-tide-thermohaline circulations (Guan, 1994). These physical conditions may lead to the increase in nutrient and thereby influence the phytoplankton size structure in the coastal region. Nano-phytoplankton were found to dominate the inner part of the ECS shelf (Fig. 8e), likely due to the increased nutrient concentrations in offshore waters resulting from the coastal region by strong convection currents. The study of Yamaguchi et al. (2012) revealed that the CDW takes approximately 2 months to move from Changjiang River mouth to the Tsushima Strait. Therefore, the nutrient supply from riverine discharge may be a major controlling factor in the large cell sizes (micro- and nano-phytoplankton) in the coastal region in the summer. These findings were consistent with the study of Jiang et al. (2015) based on field investigations who reported that the micro-sized diatoms and dinoflagellates dominated the Changjiang estuary and adjacent areas in the summer in response to available nutrients.

During autumn and winter, as wind stress strengthens and temperature decreases, convectional mixing of the water column increases and the stratification weakens, which bring nutrients upward from the underlying layer. The mixing processes through internal waves, tides and winds, as well as the terrestrial nitrate from runoff provide high nutrient condition, which promotes the phytoplankton growth and the presence of larger-sized phytoplankton, as suggested by Taylor and Joint (1990). Guo et al. (2014) also observed that the nitrate concentrations were high in coastal waters of the ECS during autumn and winter. Thus, the larger-sized phytoplankton (micro and nano) dominance was clearly observed in the coastal region (Fig. 8g-h and j-k), and high Chla were found in this region (Fig. 8C and D), consistent with previous studies by Guo et al. (2014) and Wang et al. (2014), who suggested that the most dominant phytoplankton group was chain-forming diatoms and dinoflagellates in coastal waters throughout the year.

**4.2.2 The middle shelf and shelf break of the ECS**

Similar to the coastal region, the middle shelf of the ECS exhibited the spring bloom with peak of Chla occurring in April and micro-phytoplankton dominance (Fig. 9b), mainly due to mixing process of the water column in the spring. These results agree with previous study reported by Liu et al. (2016) that during springtime, the contributions of dinoflagellates and diatoms (micro) to total Chla were relatively higher in the middle shelf region and particularly in the river plume. During summer and early autumn, due to surface warming and low wind stress, the reduced mixing and stronger thermal stratification result in less

nutrient supply to the surface layers. As reported by Guo et al. (2014), a nitracline formed in the middle shelf water in summer, and no nitracline formed in autumn and winter due to strong water mixing. This region is also affected by ocean currents carrying warm waters, e.g., the Kuroshio Branch Current to the north of Taiwan and the Yellow Sea Coastal Current (Ichikawa and Beardsley, 2002), which can enhance the water column stability. These oligotrophic conditions can favour the presence of pico-phytoplankton. Meanwhile, the coastal nutrients are transported to the middle shelf region by convection currents, but the nutrient concentrations in the middle shelf are not as high as those in the coastal region. This may be one reason that nano- and pico-phytoplankton size classes overlapped during summer and early autumn (Fig. 8e-f and Fig. 9b). In the winter, mixing of the water column increases due to strong winds (Guo et al., 2014), allowing nutrients to enter the surface layer. This condition favors the increase of micro- and nano-phytoplankton in the middle shelf of the ECS (Fig. 9b). Previous researches have shown that micro-phytoplankton dominated the shelf regions in wind-driven upwelling and mixing systems, where nutrient concentrations are high and seawater temperature are lower (Hirata et al., 2009; Sun et al., 2017; Lamont et al., 2018). These larger-sized phytoplankton dominated communities can support higher rates of photosynthesis because of their larger photosynthetic rates per unit volume (Hirata et al., 2009). Additionally, upwelling usually occurs at the shelf break of the ECS, transposing nutrients-rich waters from the subsurface layer to the upper layer (Chen et al., 2009). This condition can promote the nano-phytoplankton growth. Advective processes in the upwelling system are regarded as an important force, as well as the biochemical forces such as nutrients, in controlling the phytoplankton size structure and species composition (Smith et al., 1983). Malone (1975) reported that small nano-phytoplankton were selectively removed from upwelling regions by mass transport to the distance as a result of their low sinking rates.

### 4.2.3 The Kuroshio region and open ocean

In comparison to the ECS shelf sea, the Kuroshio region and open ocean generally exhibited relatively low chlorophyll-a concentrations (Fig. 8A-D and Fig. 9c). These phytoplankton biomass levels are controlled by a variety of forcing factors, among which the key factors are water column stability and the availability of light and nutrient (Behrenfeld, 2010; Yamaguchi et al., 2012). The Kuroshio region is largely influenced by the the Kuroshio Current, in addition to solar irradiance that governs light availability and also influence the water column stability. The mainstream of the Kuroshio Current strongly flows northeastward along around the 200 m isobath (Ichikawa and Beardsley, 2002), carrying warm and low-nutrient waters (Jiao et al., 2005). High surface temperature could strengthen the water column stability, thereby preventing the nutrient supply to the upper layer from the deeper layer (Lovelock, 2007). Thus, these oligotrophic conditions lead to the low phytoplankton biomass and promote the growth of pico-phytoplankton, which are better adapted to take advantage of such light and nutrient-

depleted conditions (Finkel et al., 2009). Liu et al. (2016) found that the importance of larger phytoplankton (diatoms and dinoflagellates) decreased appreciably in the offshore waters, and their contributions were partially replaced by small-sized phytoplankton (e.g., *Synechococcus*, *Prochlorococcus*, chrysophytes, and prymnesiophytes). On the other hand, this was also confirmed by a significant positive correlation between SST and pico-phytoplankton proportions in the KR (Fig. 10c). In

addition, temperature and salinity are implicated as important ecological determinants for some small-sized photosynthetic bacteria, e.g., *Prochlorococcus* and *Synechococcus*. *Prochlorococcus* are largely confined to the warm waters and almost absent in coastal waters in the winter (Jiao et al., 2005). Some previous studies also showed that for the abundant *Prochlorococcus* in surface waters, its lower boundaries of temperature and salinity were 15.6 and 33.5 °C in the winter respectively and 26.4 and 29.1 °C in the summer respectively (Jiao et al., 2005; Liu et al., 2016). This is particularly clear in

the Kuroshio region where pico-phytoplankton were demonian throughout the year, expect in winter and early spring when nano-phytoplankton size fractions were slight more elevated (Fig. 8 b and k; Fig. 9c). The slight increase in nano-phytoplankton proportions during winter and early spring may be related to the increased nutrient concentrations that result from vertical mixing due to stronger wind stress during this period, as reported by Liu et al. (2016) that mean surface concentrations of nutrients ($NO_3^-$ + $NO_2^-$) in the offshore Kuroshio region were higher in the winter than in the summer, and the mixed layer

depth were much deeper in the winter than in the summer due to strong vertical mixing in the winter.

## 4.3 Response of phytoplankton size class to sea surface temperature

It has previously been suggested that sea surface temperature is one of the important factors that influence the PSC dynamic (Chen, 2000; Barnes et al., 2010; IOCCG, 2014). Based on the 14-years (2003-2016) time series of the monthly SST and satellite-derived PSC data, we investigated the correlations between SST and PSC in the three subareas of the ECS (Fig. 10),

aiming to discuss the PSC response to the SST change under different hydrological conditions. In the Kuroshio region, significant negative correlation between nano-phytoplankton size fraction and SST was found ($R$= -0.66 < -0.5 and $p$< 0.001), and weak negative correlation was found for micro-phytoplankton ($R$= -0.31). Significant positive correlation between pico-phytoplankton size fraction and SST was identified ($R$= 0.64 >0.5 and $p$< 0.001) (Fig. 10c). Similarly, Chen (2000) reported that there was a significant positive correlation between *in situ* measured pico-phytoplankton proportion and water temperature.

Several studies have found that surface warming can weaken vertical mixing due to the increase in water column stability (Behrenfeld et al., 2006; Boyce et al., 2010), which causes less nutrient supply to the surface layers from the underlying nutrient-rich waters. In addition, the Kuroshio water is characterized by high salinity, high temperature, and low nutrient (Jiao et al., 2005). These oligotrophic conditions favour the presence of smaller-sized phytoplankton (pico) and restrict the growth

of larger-sized phytoplankton (micro and nano). It offers an explanation to help us for understanding the correlation between the increasing trend of SST and decreasing trend of micro- and nano-phytoplankton size fraction and increasing trend of pico-phytoplankton size fraction. Similar to the KR, a negative correlation between micro-phytoplankton proportion and SST ($R=$ -0.76 and $p< 0.001$) and a positive correlation for pico-phytoplankton ($R= 0.65$ and $p< 0.001$) were observed in the MSR (Fig. 10b). Different environmental conditions in the two subareas showed similar responses of the variability of micro- and pico-phytoplankton size fractions to SST. However, the increasing trend of SST and increasing trend of nano-phytoplankton showed a weak positive correlation ($R= 0.34$ and $p< 0.001$) (Fig. 10b), which was different to the Kuroshio region. The weak correlation suggested that nano-phytoplankton in this region may be affected by other factors (e.g., grazing-nitrogen rate) other than SST (Furuya et al., 2003). For instance, Barlow et al. (2016) reported that nano-phytoplankton (e.g, flagellates) were dominant in warmer shelf region, because they are better utilising the increase in nutrient concentrations after upwelled water has warmed. In the mouth area of Changjiang River, the SST were negatively ($R= -0.59$ and $p< 0.001$), positively ($R= 0.58$ and $p< 0.001$), and positively ($R= 0.54$ and $p< 0.001$) with micro-, nano-, and pico-phytoplankton size fraction, respectively (Fig. 10a). The water body in coastal region mixes well in winter with low SST and has a weak stratification of water column in summer with high SST, as the hydrodynamic in coastal water is dominated by the variation of wind-tide-thermohaline circulations (Guan, 1994). In some degree, increasing of SST could result in the decrease of larger-sized phytoplankton (micro and nano) and the increase of small-sized phytoplankton (pico). However, the trend of rising SST and the increasing nano-phytoplankton size fraction in the MCJR were observed. This may be related to the optimum temperature for the growth of different algal groups. Additionally, pervious studies have shown that the nutrient structure in the ECS have altered by the Changjiang discharge, especially for the Changjiang estuary and adjacent area (Zhang et al., 2007; Wang et al., 2014). The change of nutrient structure (increase in N/P ratio) may play an important role in regulating the phytoplankton community structure in the MCJR (Guo et al., 2014). These results suggested the interannual variability of PSC in coastal waters is more complicated than in offshore waters. The detailed study focusing on the mechanism of the PCS change in the ECS is still required.

Overall, the correlations between PSC and SST (Fig. 10) indicated SST is an important factor influencing the PSC dynamic in the ECS. The interannual variations of phytoplankton size classes in the ECS were complicated and could not be fully explained by the individual factor. Further investigations therefore are required to understand the interannual variability of the PSC in the ECS and its response to environmental factors, e.g., wind speed, riverine discharge, and monsoon forcing.

**5 Conclusions**

In this study, the PSC model was regionally tuned for application to the ECS using extensive *in situ* measured data covering

various seasons and environmental conditions in the ECS. When the refined model was applied to MODIS observations, there was a critical step to reconstruct satellite remote sensing reflectance at blue wavebands. It led to reliable performance of the refined PSC model on MODIS observation, which showed good agreement with the HPLC-derived PSC results, with almost all of the samples falling within ±20% fraction range. Along the way, our present study preliminarily estimated spatial distributions of the PSC in the ECS from space. The refined PSC model was applied to satellite data from MODIS during 2003 and 2016 to investigate the PSC distribution at seasonal scale. The obtained results showed that the PSC in the ECS varied across both spatial and temporal scales. The seasonality of the PSC in the ECS was likely to be related to the vertical structure of the water column, upwelling, sea surface temperature, and the Kuroshio Current. It was also affected by riverine discharge and human activity, especially for coastal waters. The interannual and longer-term variations in phytoplankton size class in the East China Sea and their mechanisms are needed to be investigate in the future.

**Author Contribution**

H. L. Zhang analysed the data and wrote the manuscript; S. Q. Wang and Z. F. Qiu contributed to design this study and interpretation of the results; D. Y. Sun and J. Ishizaka revised the draft; S. J. Sun and Y. J. He provided comments and suggestions to improve the manuscript.

**Conflict of Interest**

The authors declare that there has no conflict of interest.

**Acknowledgements**

This research was jointly supported by the National Key Research and Development Program of China (2016YFC1400904), the National Natural Science Foundation of China (41506200, 41576172, and 41276186), the Natural Science Foundation of the Jiangsu Higher Education Institutions of China (15KJB170015), the Provincial Natural Science Foundation of Jiangsu in China (BK20150914, BK20151526, and BK20161532), the National Program on Global Change and Air-sea Interaction (GASI-03-03-01-01), the Public Science and Technology Research Funds Projects of Ocean (201005030), a project funded by "the Priority Academic Program Development of Jiangsu Higher Education Institutions (PAPD)", the Research and Innovation Project for College Graduates of Jiangsu Province (KYLX16_0952), and the China Scholarship Council.

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

# Tables

**Table 1.** Parameter $\beta_i$ values for the PSC model development.

| Class | $N$ | $R$ | RMSE | $\beta_0$ | $\beta_1$ | $\beta_2$ | $\beta_3$ | $\beta_4$ |
|-------|-----|-----|------|-----------|-----------|-----------|-----------|-----------|
| Micro | 170 | 0.89 | 0.11 | 1.05 | 3.48 | -4.34 | -13.09 | 16.04 |
| Nano | 170 | 0.70 | 0.11 | - | - | - | - | - |
| Pico | 170 | 0.84 | 0.11 | -2.56 | -1.52 | 1.24 | -25.87 | -1.86 |

**Table 2.** Results of the matchup comparison of satellite $R_{rs}(\lambda)$ with *in situ* measurements.

| | Wavelength (nm) | | | | | | | | | |
|---|---|---|---|---|---|---|---|---|---|---|
| | **412** | **443** | **469** | **488** | **531** | **547** | **555** | **645** | **667** | **678** |
| *N* | 21 | 21 | 21 | 21 | 21 | 21 | 21 | 21 | 21 | 21 |
| *R* | 0.46 | 0.73 | 0.85 | 0.88 | 0.95 | 0.96 | 0.97 | 0.90 | 0.86 | 0.85 |
| RMSE | 0.0026 | 0.0019 | 0.0016 | 0.0016 | 0.0011 | 0.0011 | 0.0012 | 0.00081 | 0.00077 | 0.00079 |
| MAPE(%) | 47.33 | 36.90 | 27.25 | 19.92 | 16.39 | 14.90 | 18.41 | 54.86 | 91.39 | 111.3 |

**Table 3.** Statistical parameters and coefficients for the algorithm to reconstruct $R_{rs}$ data.

| wavelengths | $N$ | $R^2$ | RMSE | MAPE | Constant coefficients $K_j, j=0,1,\ldots n$ |
|---|---|---|---|---|---|
| 412 nm | 341 | 0.99 | $6.3*10^{-4}$ | 8.50% | $4.43*10^{-4}$, 3.91, -3.19, 0.20, 0.72, -0.69 |
| 443 nm | 341 | 0.99 | $2.2*10^{-4}$ | 3.13% | $7.39*10^{-5}$, 2.50, -1.59, -0.36, 1.22, -0.77 |

**Table 4.** Comparison of the original satellite $R_{rs}$ and reconstructed $R_{rs}^{rc}$ with *in situ* measured data at 412 and 443 nm.

| wavelength | N | original satellite $R_{rs}$ | | | reconstructed satellite $R_{rs}^{rc}$ | | |
|---|---|---|---|---|---|---|---|
| | | R | RMSE | MAPE(%) | R | RMSE | MAPE(%) |
| 412 nm | 21 | 0.46 | 0.0026 | 47.33 | 0.70 | 0.0019 | 35.15 |
| 443 nm | 21 | 0.73 | 0.0019 | 36.90 | 0.80 | 0.0017 | 34.53 |

**Table 5.** Results of the matchup comparison for Fig. 6.

| wavelength | $N$ | satellite-derived $a_{ph}^{rc}$ | | | | satellite-derived $a_{ph}$ | | | |
|---|---|---|---|---|---|---|---|---|---|
| | | $R$ | RMSE | MAPE(%) | PVP (%) | $R$ | RMSE | MAPE(%) | PVP (%) |
| 412 nm | 22 | 0.80 | 0.27 | 15.35 | 100 | 0.58 | 0.56 | 58.15 | 72.73 |
| 443 nm | 22 | 0.83 | 0.23 | 14.87 | 100 | 0.48 | 0.67 | 63.95 | 72.73 |
| 469 nm | 22 | 0.85 | 0.21 | 11.62 | 100 | 0.62 | 0.55 | 48.35 | 81.82 |
| 488 nm | 22 | 0.84 | 0.22 | 11.52 | 100 | 0.78 | 0.37 | 31.93 | 90.91 |
| 531 nm | 22 | 0.80 | 0.31 | 20.82 | 86.36 | 0.86 | 0.73 | 45.57 | 54.55 |
| 547 nm | 22 | 0.69 | 0.43 | 20.38 | 90.91 | 0.69 | 0.52 | 35.71 | 63.64 |

# Figures

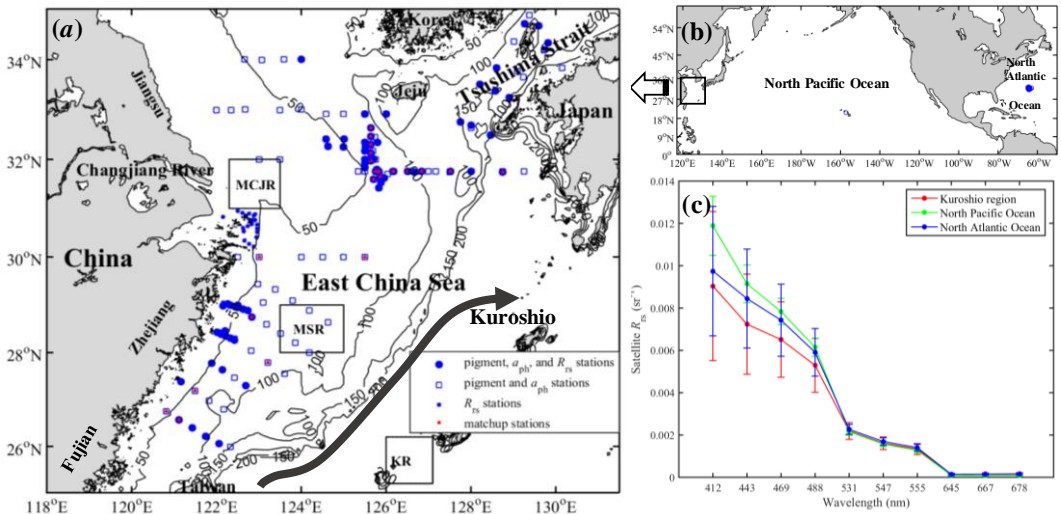

5   **Fig. 1**. Distribution of *in situ* and matchup dataset and locations of the selected subareas (black boxes) (*a*), namely MCJR (mouth area of Changjiang river), MSR (middle shelf region), and KR (Kuroshio region); locations of sampling stations collected in the North Pacific and North Atlantic oceans from the NASA SeaBASS archive (b); the average satellite $R_{rs}(\lambda)$ spectra from 2003 to 2016 in the Kuroshio region, North Pacific ocean, and North Atlantic ocean (blue circles in b) (c). Error bars represent standard deviations of the means.

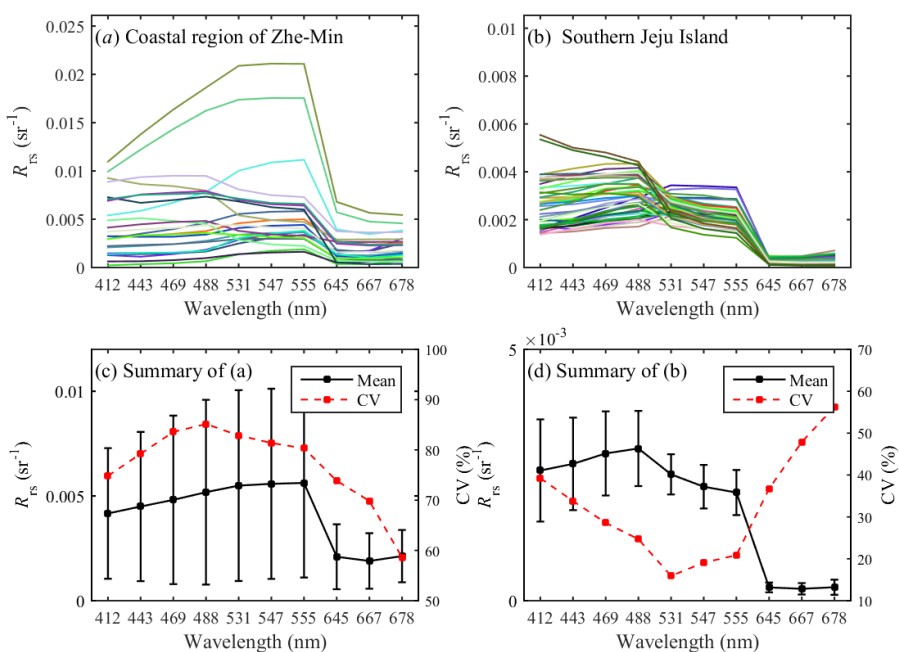

**Fig. 2**. $R_{rs}(\lambda)$ spectra at MODIS wavelengths collected in the coastal region of Zhe-Min (a) and southern Jeju Island (b); Mean spectra and coefficient of variation (CV) of $R_{rs}(\lambda)$ in the coastal region of Zhe-Min (c) and southern Jeju Island (d). The CV is derived as the standard deviation (SD) over the mean.

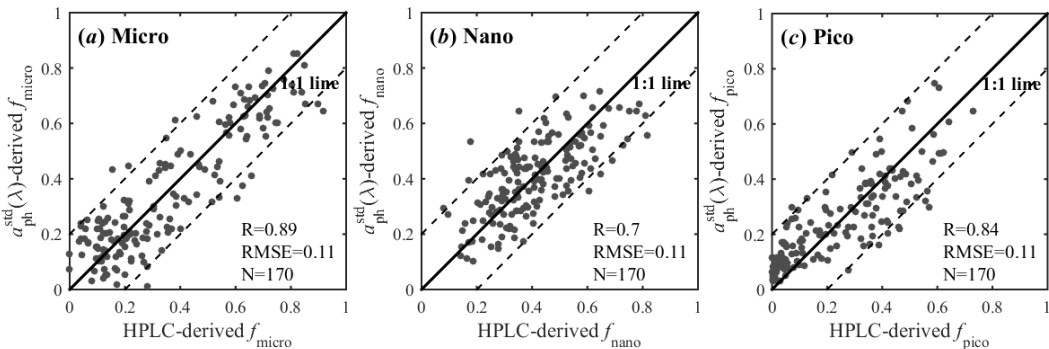

**Fig. 3.** Comparison between *in situ* $a_{ph}^{std}(\lambda)$-derived and HPLC-derived PSC for micro- (a), nano- (b), and pico-phytoplankton (c). Dashed lines represent the ± 20% fraction range relative to the 1:1 line.

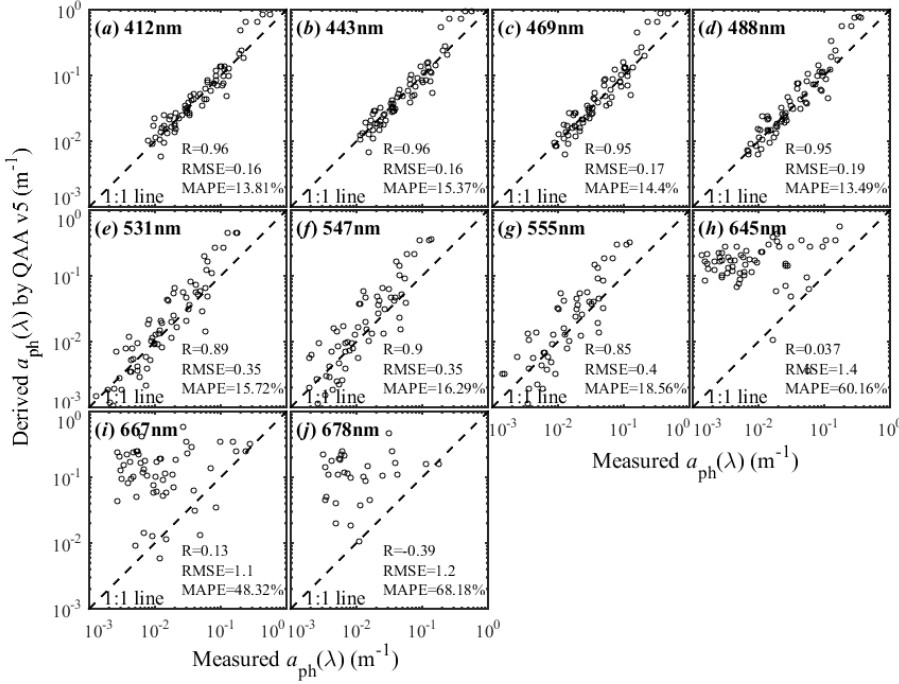

**Fig. 4.** Comparison of $a_{ph}$ derived from $R_{rs}$ using QAA_v5 with *in situ* measured $a_{ph}$ data.

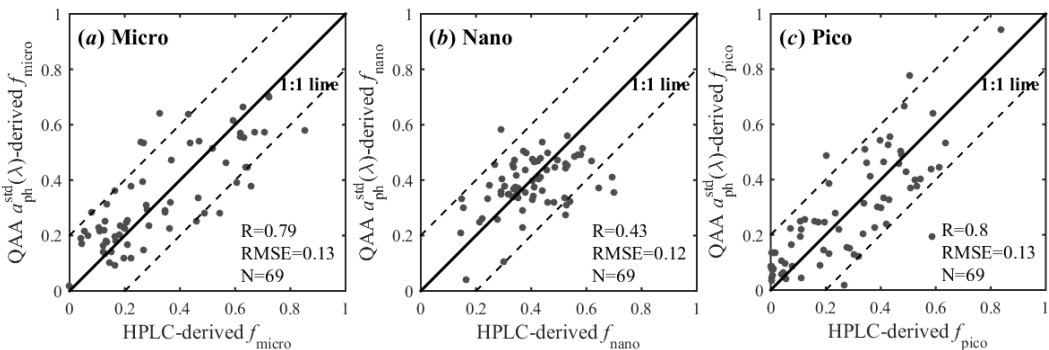

**Fig. 5.** Comparisons of the PSC modeled using $a_{ph}$ derived from $R_{rs}$ with the HPLC-derived values for micro- (*a*), nano- (b),

and pico-phytoplankton (c).

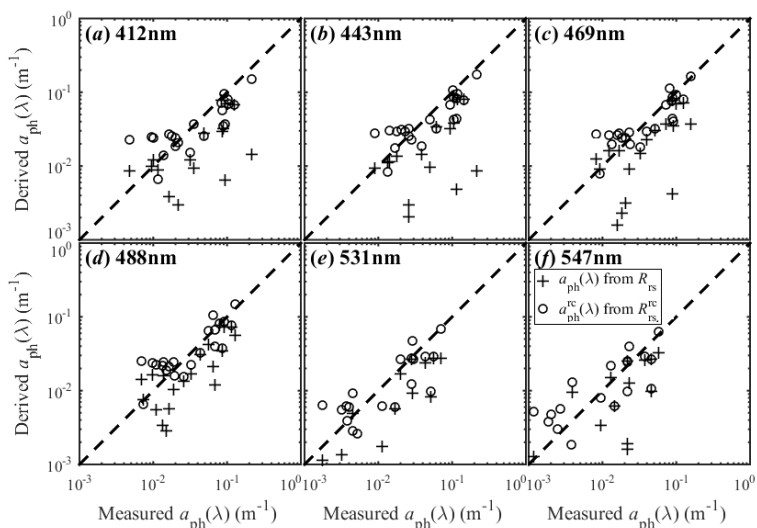

**Fig. 6**. Comparison of the satellite-derived $a_{ph}$ (crosses) and satellite-derived $a_{ph}^{rc}$ (open circles) with *in situ* measured data at 412, 443, 469, 488, 531, and 547 nm.

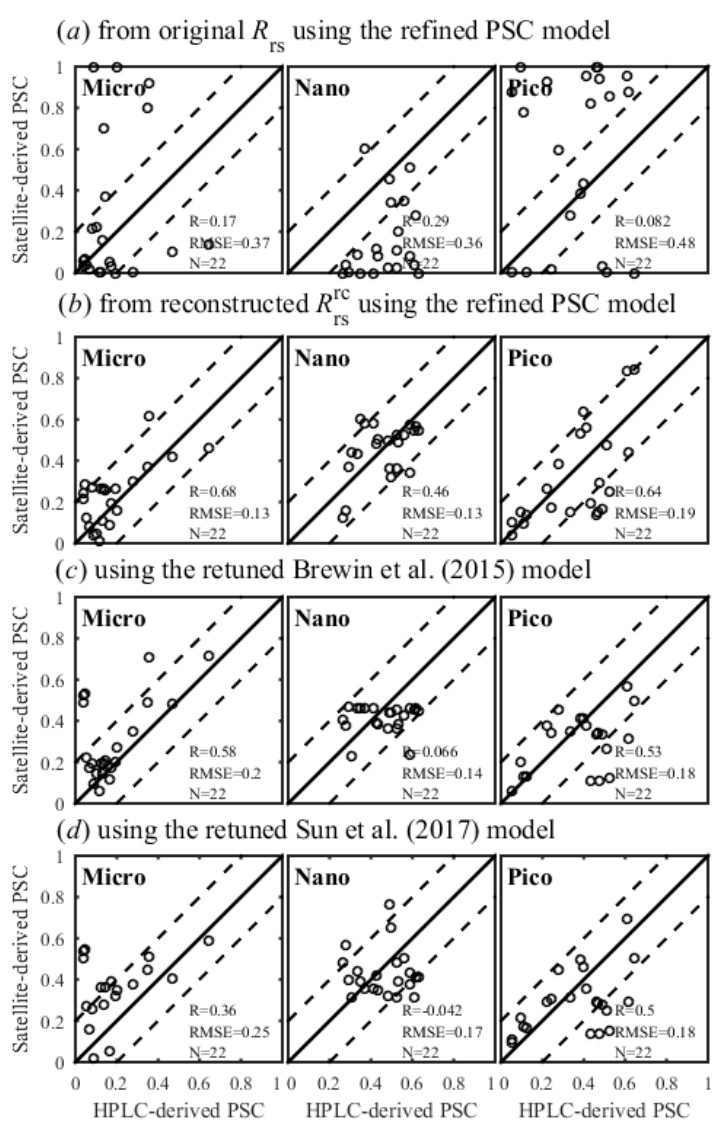

**Fig. 7.** Comparison of the HPLC-derived PSC with the satellite-derived PSC values from the original satellite $R_{rs}$ (a) and the reconstructed satellite $R_{rs}^{rc}$ (b) using the refined PSC model in this study; using the retuned Brewin et al. (2015) model (c); using the retuned Sun et al. (2017) model (d). Solid lines denote the 1:1 lines.

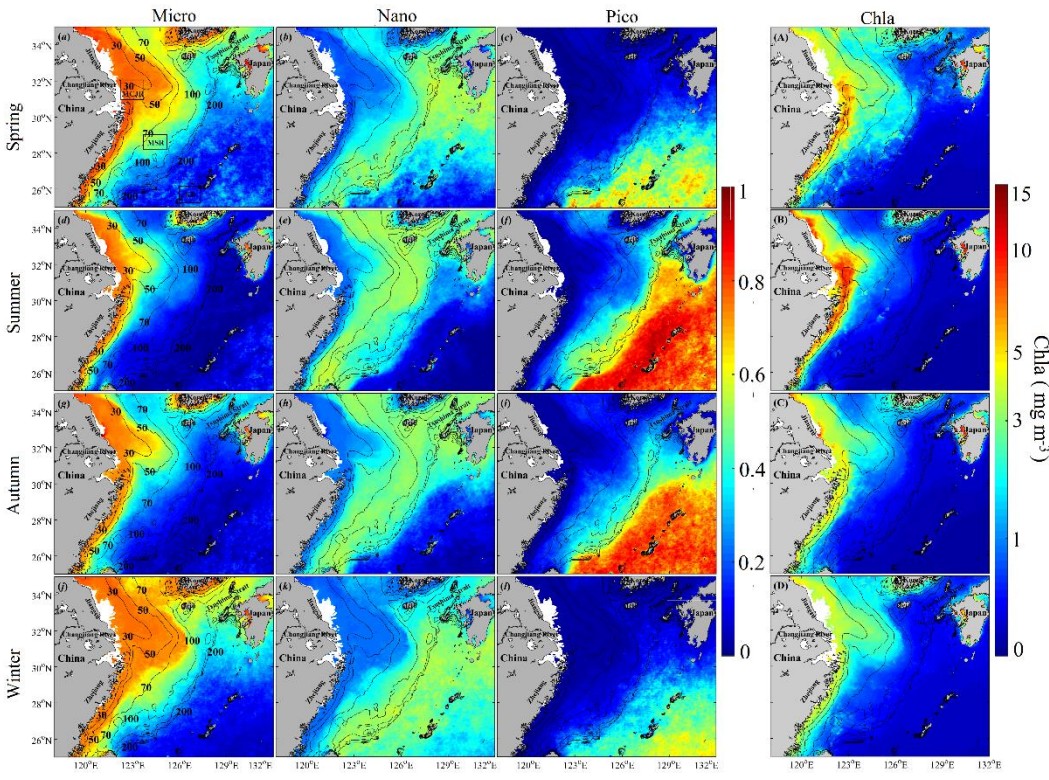

**Fig. 8.** Seasonal distributions of the PSC (*a*-l) and Chla (A-D, right panel) in the ECS during 2003-2016.

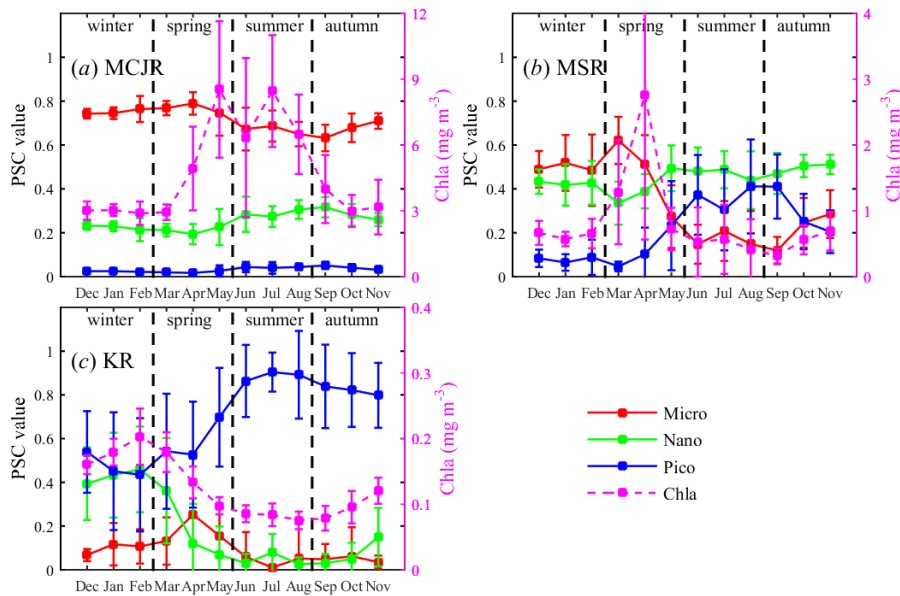

**Fig. 9.** Monthly climatological PSC and Chla from 2003 to 2016 in the mouth area of Changjiang river (MCJR) (*a*), middle shelf region (MSR) (b), and the Kuroshio region (KR) (c). Error bars indicate standard deviations of the means.

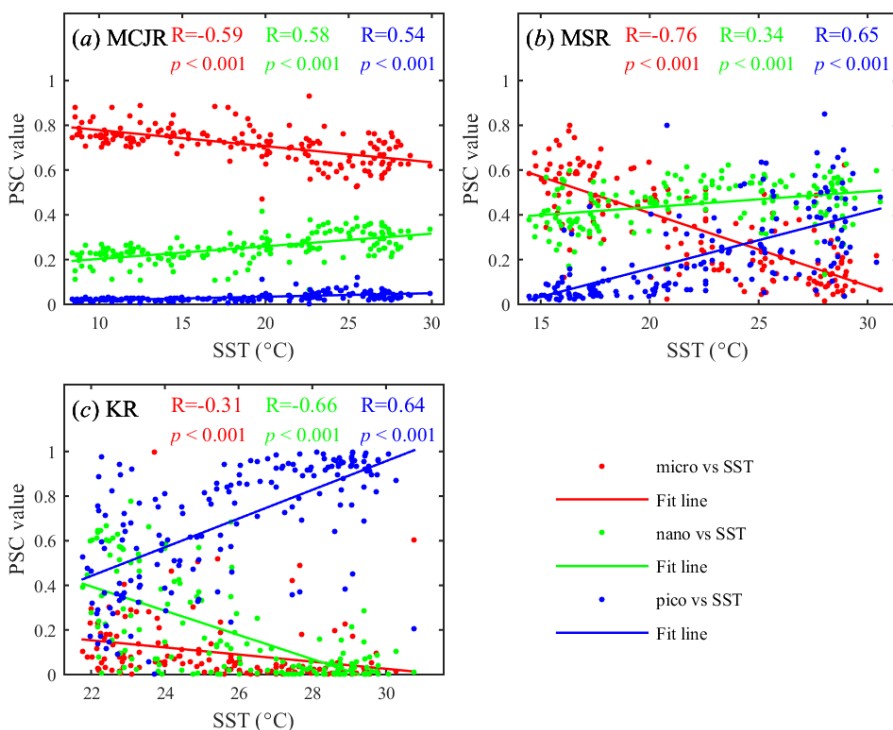

**Fig. 10.** The scatterplots showing the relationships between the monthly phytoplankton size fractions and SST from 2003 to 2016 for the MCJR (a), MSR (b), and KR (c).