# Peer review of "Phytoplankton size class in the East China Sea derived from MODIS satellite data"

_Biogeosciences, 2017_

## Referee Comment (RC1) · Anonymous Referee #1 · 25 Feb 2018

The East China Sea is a marginal sea that includes a large area of shallow continental shelf. Water dynamics in the ECS is very complex due to the influence of wind, intrusion water of the Kuroshio Current, the freshwater discharge from the Changjiang (Yangtze) River, and the topography. Phytoplankton productivity and community structures show large gradients from the open ocean to the shelf and to the estuarine waters. In this study, the spectral-based model proposed by Wang et al. (2014) was modified to retrieve the phytoplankton size classes (PSCs) from the MODIS derived phytoplankton absorption spectra. This is the main novelty of this paper, which is also the important basis for studying the spatial and seasonal variability of PSCs from MODIS ocean color data. As shown by Wang et al. (2014), based on in situ measured phytoplankton absorption spectra(aph($\lambda$)), or aph($\lambda$) derived from in situ measured remote sensing
reflectance (Rrs) with QAA algorithm, this approach showed good performances. In this paper, uncertainty of aph spectra retrieved directly from MODIS ocean color data is key for determining the accuracy of this model. Validation of this model based on about 21 data points is encouraging, however, as shown in Fig. 7, there are still large deviations between measured and estimated value. The general spatial distribution of PSCs maybe reasonable. But for some specific areas the credibility of these results is still an open question. Discussion (4.2) about the spatial distribution and the seasonal variability of phytoplankton size classes are mostly descriptive and relatively superficial. It's a good point to pick up three specific areas to discuss the seasonal variability and its relationship with SST. More explanation and further discussion about the physical and chemical environment for these areas could be useful for readers to know the feasibility of this remote sensing model. I noticed that a paper entitled "Remote-Sensing Estimation of Phytoplankton Size Classes From GOCI Satellite Measurements in Bohai Sea and Yellow Sea" was published recently in JGR by the same group. They also showed some results about this area. However, these results seem to be a little different from each other. I recommended authors to do more work about the validation and comparison of this model.

Some specific questions or recommendations as follows: (1) In ECS, the "abundance-based" approach may not perform as well as that in open ocean. How about the general variability of PSCs(fractions) with the total Chl-a according to in situ data? Since distribution of phytoplankton biomass may help us to explain the spatial variability of PSCs, I also recommend to add the seasonal distribution of total Chl-a in Fig.8. (2) For processing MODIS data, which algorithm was used for estimating the total Chl-a? How about the validation results with in situ match-up data points? (3) About the reconstruction of Rrs at 412 and 443nm wavebands, more data from SeaBass dataset were used for developing the relationship. Does this relationship exhibit the same distribution over coastal waters and open ocean? These coefficients (K) could be shown in a Table. (4) As shown in Section 4.2, the spring bloom was found to occur frequently in the mouth area of Changjiang river and middle shelf region. How about the performance

of this "spectral-based" model for PSCs retrieval for bloom waters? Does it give better results than the others ("abundance-based" model)? Clear comparison and discussion about this point could be very helpful for supporting the credibility of this model. (5) Results shown in Fig.9 is very interesting, which have already attracted much attentions from marine ecologists. I recommend authors to do further discussion about these variabilities referring those published results. At the same time, those published results about the spatial and seasonal variations of PSCs in ECS could be used for validating the MODIS derived values. (6) Temperature itself is an important factor governing the distribution of phytoplankton, which also provides a quantitative index of the physio-chemical state of the marine environment. How about the correlation between total Chl-a (phytoplankton biomass) with SST? As shown in Table 5, these correlation between SST and size fractions may have different underlying mechanism for the 3 different subareas. Some more explanations about the hydrological backgrounds of these subareas are expected to deepen the understanding.

Some specific technical suggestions: 1. Fig.1, Mean Rrs spectra for coastal waters of ECS could be helpful for reader to know exactly the ocean color variability in ECS (which covers many water types). 2. Coefficients of K in Equation (9) could be shown in a Table. 3. Fig.8, add the spatial distribution of total Chla for 4 seasons. 4. Show locations for the three subareas in Fig.8 and introduce the box size in Data and Methods. 5. Fig.9, enlarge the y-axis of (c) and (d) for total Chl-a for clarity. 6. For results shown in Table 5, a figure showing time series of size fractions and SST may be helpful for discussing their correlations.

―――――――――――――――――――――――――――――

---

## Referee Comment (RC2) · Anonymous Referee #2 · 28 Feb 2018

The authors used in situ datasets and reconstructed Rrs data from MODIS to estimate the PSC in the East China Sea and investigated the seasonal variability of the PSC in the ECS using ∼14 years MODIS-Rrs derived PSC data. The authors tuned the PCA approach proposed in an earlier study by Wang et al. 2014 to derive PSC from absorption measurements. The tuned approach was also applied to MODIS data via reconstructing MODIS Rrs in the blue bands and the QAA inversion method for MODIS-derived phytoplankton absorption. Improvements in methods led to better retrievals in this region, which is encouraging. Seasonality of PSC was also investigated by discussing the relating factors such as water vertical structure, temperature, upwelling, etc., providing a better understanding on the PSC in this region with regard to environmental changes. The manuscript was overall well written despite a few inconsistencies

in the tense and wording. The authors have also improved the manuscript according to what I suggested in the first review, including selecting a few typical subregions and analyzing the climatological variation besides the seasonality.

My further suggestions are as below and details were highlighted in the manuscript. 1 As phytoplankton absorption and Chla are quite important in PSC estimation, probably the authors can also display the seasonal distributions of QAA derived aph (for example aph(440) as aph(675) is not successfully estimated) and QAA derived aph from reconstructed MODIS Rrs in the ECS? As QAA only retrieves IOPs but not Chla, this can at least give a hint on how the Chla distributes and changes over seasons by showing aph distributions. I then noticed that MODIS Chla products were also used. How is the MODIS Chla compared with the in situ ones at matchups? and how was the MODIS derived aph versus in situ Chla? compared to the MODIS chla, does QAA aph(440) had a better correlation with in situ Chla? 2 The authors stated that the findings presented here complement and enhance recent studies that have demonstrated that satellite ocean color data can be used to retrieve the PSC in the ECS. What other studies in this region or in China's seas? How are your results compared to these studies? are they basically in consistence with the others? 3 Description on statistics sometimes is not very precise. Such as the authors used 'acceptable errors' or 'significant correlation' but did not explain how you defined acceptable or significant, maybe P values help a bit or change the way of interpretation. 4 The description of the sub-regions is inadequate, please specify their sizes, and use boxes to specify the exact size and location in the map. 5 When discussing the PSC response to the SST, it might be also helpful to also show the SST variations.

More detailed comments and technical corrections were listed in the attached file.

Please also note the supplement to this comment:
https://www.biogeosciences-discuss.net/bg-2017-508/bg-2017-508-RC2-supplement.pdf

[Figure]

**Supplement:**

[revised manuscript text omitted]

---

## Author Response (AR1)

Dear Reviewers,

We deeply appreciate your comments and effort towards improving our manuscript. We have taken your constructive comments carefully in the revision of our manuscript. For the revision, please kindly refer to the point-to-point responses as followings and the revised manuscript. The changes we made have been noted in the blue color for highlighting.

**Response to Reviewer #1**

The East China Sea is a marginal sea that includes a large area of shallow continental shelf. Water dynamics in the ECS is very complex due to the influence of wind, intrusion water of the Kuroshio Current, the freshwater discharge from the Changjiang (Yangtze) River, and the topography. Phytoplankton productivity and community structures show large gradients from the open ocean to the shelf and to the estuarine waters. In this study, the spectral-based model proposed by Wang et al. (2014) was modified to retrieve the phytoplankton size classes (PSCs) from the MODIS derived phytoplankton absorption spectra. This is the main novelty of this paper, which is also the important basis for studying the spatial and seasonal variability of PSCs from MODIS ocean color data. As shown by Wang et al. (2014), based on in situ measured phytoplankton absorption spectra ($a_{ph}(\lambda)$), or $a_{ph}(\lambda)$ derived from in situ measured remote sensing reflectance ($R_{rs}$) with QAA algorithm, this approach showed good performances. In this paper, uncertainty of $a_{ph}$ spectra retrieved directly from MODIS ocean color data is key for determining the accuracy of this model. Validation of this model based on about 21 data points is encouraging, however, as shown in Fig. 7, there are still large deviations between measured and estimated value. The general spatial distribution of PSCs maybe reasonable. But for some specific areas the credibility of these results is still an open question.

Response: Thank you for your comments. We agree with you that there were still deviations between measured and derived result, while it should be admitted that validation using the satellite matchups (*N*=22) showed generally reasonable model performance when we use the reconstructed satellite $R_{rs}$ (Fig. R1b); meanwhile this performance was significantly better than that using original satellite data (see Fig. R1a).

In addition, to further examine the performance of the refined PSC model in our study, our refined PSC model was compared with other two published PSC models (i.e., the Brewin et al. (2015) model and the Sun et al. (2017) model). The scatter distributions of satellite-derived PSC using our refined PSC model were closer to the 1:1 line than those of the other two published models. According to the statistical indicators, our refined PSC model had the best performance, with the R values of 0.68, 0.46, and 0.64 and RMSE values of 0.13, 0.13, and 0.19 for micro-, nano-, and pico- phytoplankton, respectively (Fig. R1b). For the Brewin et al. (2015) model (Fig. R1c), the R values of 0.57, 0.093, and 0.52 and RMSE values of 0.26, 0.15, and 0.21 were observed for micro-, nano-, and pico- phytoplankton, respectively. For the Sun et al. (2017) model (Fig. R1d), the R values were 0.22, 0.099, 0.37 for micro-, nano, and pico-phytoplankton, when the corresponding RMSE values were 0.17, 0.18, and 0.19, respectively. The Brewin et al. (2015) model and the Sun et al. (2017) model had relatively poor performance in the ECS, especially for micro- and pico-phytoplankton. These results of Fig. R1 indicated that the performance of our refined PSC model using the reconstructed satellite data were better than those of the Brewin et al. (2015) model and the Sun et al. (2017) model in our study region. In the revised manuscript, we have added the explanations regarding this issue in Section 3.4.

Therefore, we have investigated the spatiotemporal variability of the PSC in the ECS based on satellite-derived products from the reconstructed satellite data, and the general spatial distribution of PSC was reasonable. Fig. R2 showed the comparison between the spatial distributions of PSC during summer and autumn derived from both MODIS and field measurements. Overall, their general distributions patterns agreed with each other. For micro-phytoplankton, high values were generally found in near-shore regions with lower values in offshore waters during summer and autumn. For nano-phytoplankton, both MODIS and field measurements showed high values in the middle shelf region of the ECS. For pico-phytoplankton, both MODIS and field measurements showed low values in the coastal region during summer and autumn and high values in the coastal waters of western Japan during summer. However, it is a pity that there was no satellite matchup in the Kuroshio waters of the ECS to assess the credibility of satellite-derived PSC results in this region as you concerned. In future research, we will add additional field data to the PSC model developing and to validation of the credibility of satellite-derived PSC results, especially for the Kuroshio region of the ECS.

[Figure]

Fig. R1 Comparison of the HPLC-derived and satellite-derived PSC result from the original satellite $R_{rs}$ using the refined PSC model in this study(a); from the reconstructed satellite $R_{rs}$ using the refined PSC model in this study (b); using the Brewin et al. (2015) method (c); using the Sun et al. (2017) model (d). Solid lines denote the 1:1 lines.

[Figure]

Fig. R2. Comparison of spatial distributions of PSC in the ECS between satellite retrievals and field measurements during summer and autumn.

Discussion (4.2) about the spatial distribution and the seasonal variability of phytoplankton size classes are mostly descriptive and relatively superficial. It's a good point to pick up three specific areas to discuss the seasonal variability and its relationship with SST. More explanation and further discussion about the physical and chemical environment for these areas could be useful for readers to know the feasibility of this remote sensing model.

Response: Thank you for your valuable suggestions. In the revised manuscript, based on the different seasonal distributions of the PSC and Chla in the ECS sub-regions (as shown in Fig. 8 and Fig. 9 in the revised manuscript), we have discussed the regional scale characterization of the full seasonal cycle in stellate-derived PSC. Meanwhile, more explanation and discussion regarding the related physical and biochemical effects were added for helping to understand the spatiotemporal variability of the PSC in the ECS. Full details are in Discussion, Section 4.2 (Section 4.2.1 - The coastal region; Section 4.2.2 - The middle shelf and shelf break of the ECS; Section 4.2.3 - The Kuroshio region and open ocean) in the revised manuscript. Thank you.

I noticed that a paper entitled "Remote-Sensing Estimation of Phytoplankton Size Classes From GOCI Satellite Measurements in Bohai Sea and Yellow Sea" was published recently in JGR by the same group. They also showed some results about this area. However, these results seem to be a little different from each other. I recommended authors to do more work about the validation and comparison of this model.

Response: Thank you for your valuable suggestions. Based on 22 satellite matchup samples, we have estimated the satellite-derived PSC results using the refined model in our study and using the Sun et al. (2017) model with its original parameters, respectively. These satellite-derived PSC results were validated using HPLC-derived values, in order to assess and compare the performance of the two PSC models in the ECS (Fig. R3). For the Sun et al. (2017) model, the $R$ values were 0.22, 0.099, 0.37 for micro-, nano, and pico-phytoplankton, with the corresponding RMSE values of 0.17, 0.18, and 0.19, respectively. For our refined PSC model, the $R$ values of 0.68, 0.46, and 0.64 and RMSE values of 0.13, 0.13, and 0.19 were observed for micro-, nano-, and pico- phytoplankton, respectively. These results indicated that the performance of our refined PSC model in the ECS was better than the Sun et al. (2017) model, especially for micro- and pico-phytoplankton. The relatively poor performance of the Sun et al. (2017) model in the ECS may be due to the limited applicability of this model, since this model was

actually developed based on the field data collected from the Bohai and Yellow seas where the waters are relatively more turbid than those in the ECS.

[Figure]

Fig. R3 Comparison of the HPLC-derived and satellite-derived PSC results from the reconstructed satellite data using the refined PSC model in this study (a); using the Sun et al. (2017) model (b).

Some specific questions or recommendations as follows:

(1) In ECS, the "abundance based" approach may not perform as well as that in open ocean. How about the general variability of PSCs (fractions) with the total Chl-a according to in situ data? Since distribution of phytoplankton biomass may help us to explain the spatial variability of PSCs, I also recommend to add the seasonal distribution of total Chl-a in Fig. 8.

Response: Thank you for your comments and suggestions. The scatterplots between HPLC-derived PSC and measured Chla were drawn to show the variability of PSCs with the total Chla for *in situ* data (Fig. R4). Meanwhile, we have also plotted the HPLC-derived PSC as a function of the *in situ* total Chla with the original coefficients of Brewin et al. (2015) for the global ocean. It can be clearly seen that the data were quite scattered, and the fractional variation of each population in the ECS as a function of Chla don't strictly agree with the results of Brewin et al. (2015) (black curves in Fig. R4), thus the "abundance based" approach may not perform well in the East China Sea as suggested by Wang et al. (2014). However, in general, micro-, nano-, pico-phytoplankton dominate at high, intermediate, and low Chla, respectively. Therefore, as you suggested, we have added the seasonal distribution of total Chl-a in Fig.

8 (Fig. R5 in this response) in the revised manuscript, for helping us to explain the spatial variability of PSCs in our study region. These seasonal Chla were generated by averaging the standard monthly Chla products from the MODIS-Aqua sensor provided by the NASA Ocean Color website. In the revised manuscript, we have added the explanations regarding this issue in Section 3.5.

[Figure]

Fig. R4. Scatterplots showing the variability of the size fractions of (a) micro, (b) nano, and (c) pico-phytoplankton with the *in situ* measured Chla. The black curves denote the PSC as a function of the total Chla with its original coefficients of Brewin et al. (2015) for the global ocean.

[Figure]

10      Fig. R5. Seasonal distributions of the PSC (*a - l*) and Chla (A-D, right panel) in the ECS during 2003-2016.

(2) For processing MODIS data, which algorithm was used for estimating the total Chl-a? How about the validation results with in situ match-up data points?

Response: Thank you for your question. In this study, we used the standard monthly Chla products of

MODIS-Aqua sensor provided by the NASA Ocean Color website. These standard Chl-a products were obtained using the combined algorithm of the O'Reilly band ratio OCx algorithm and the Hu color index (CI) algorithm (see details in Hu et al. (2012)).

To assess the accuracy of the MODIS Chl-a data, we validated the satellite Chla data using *in situ* measured Chl-a based on 22 matchups (Fig. R6). The satellite Chla data in the matchup dataset were obtained from the daily Level 2 Chla products from MODIS Aqua sensor. This matchup dataset only consisted of satellite Chla with an overpass time window within 5h before and after field data. To avoid the effects of outliers, the median Chla values for a $3 \times 3$ pixels window centered on the locations of the sampling stations were defined as satellite Chla. As shown in Fig. R5, satellite Chl-a generally agreed well with *in situ* measured Chl-a, with $R^2$, RMSE and MAPE values of 0.85, 0.16 mg·m$^{-3}$ and 31.38%, respectively. These results suggested that the satellite Chl-a had good accuracy in our study region, which was considered generally acceptable in remote sensing research (Gregg and Casey, 2004; Siswanto et al., 2011). Therefore, we have used the standard MODIS Chla products to help us better understand the spatiotemporal variability of PSC in the ECS.

[Figure]

Fig. R6. Comparison of the satellite Chl-a with *in situ* measured values.

(3) About the reconstruction of $R_{rs}$ at 412 and 443nm wavebands, more data from SeaBass dataset were used for developing the relationship. Does this relationship exhibit the same distribution over coastal waters and open ocean? These coefficients ($K$) could be shown in a Table.

Response: Thank you for your comments. Our study region includes coastal region and open ocean. In our study, the same sets of coefficients for reconstruction $R_{rs}$ at 412 and 443 nm were used in both coastal region and open ocean, although there were slight differences between the coefficients for different datasets (i.e., the coastal region dataset, the open ocean dataset, and the combined dataset) (as shown in Table R1). Although the same sets of coefficients were used in our study region, the obtained results of reconstruction $R_{rs}$ showed satisfactory performance, and yielded reasonable distributions of satellite-derived PSC in our study region (see Fig. 6, Fig. 7 and Fig. 8 in the revised manuscript). Indeed, if the different coefficients of algorithms for reconstruction $R_{rs}$ are retrieved for the coastal region and open ocean respectively, the performance of reconstruction satellite data may be improved. However, along the way, the water in our study region are discriminated to the two types (i.e., coastal water and open ocean waters) before the class-based reconstruction algorithms are applied, and thus development of optical criteria to discriminate water types is needed. These processes increase the complexity of the method. Therefore, weighing the operation convenience and accuracy of the method, in our study, we have used the same sets of coefficients ($K$) in the entire study area (see Table 3 in the revised manuscript). Meanwhile, Fig. 5 in the manuscript has been removed. Thank you.

Table R1 Comparison of equation coefficients of reconstruction $R_{rs}$ for three different datasets.

| wavelengths | Dataset | $K_0$ | $K_1$ | $K_2$ | $K_3$ | $K_4$ | $K_5$ |
|---|---|---|---|---|---|---|---|
| | Combined dataset | $4.43*10^{-4}$ | 3.91 | -3.19 | 0.20 | 0.72 | -0.69 |
| 412 nm | Coastal region | $2.09*10^{-4}$ | 2.07 | -1.49 | 0.27 | 0.46 | -0.47 |
| | Open ocean | $4.43*10^{-4}$ | 5.01 | -5.05 | 1.19 | 1.17 | -0.62 |
| | Combined dataset | $7.39*10^{-5}$ | 2.50 | -1.59 | -0.361 | 1.22 | -0.77 |
| 443 nm | Coastal region | $4.48*10^{-5}$ | 1.93 | -0.93 | -0.24 | 0.26 | -0.07 |
| | Open ocean | $7.26*10^{-5}$ | 2.24 | -1.11 | -0.69 | 2.42 | -2.29 |

(4) As shown in Section 4.2, the spring bloom was found to occur frequently in the mouth area of Changjiang river and middle shelf region. How about the performance of this "spectral-based" model for PSCs retrieval for bloom waters? Does it give better results than the others ("abundance-based" model)? Clear comparison and discussion about this point could be very helpful for supporting the credibility of this model.

Response: Thank you for your valuable suggestions. As you suggested, we have chosen the the Brewin et al. (2015) model as a "abundance-based" model to compare with our PSC model. First, based on 22 satellite matchups, the performance of our refined PSC model was compared with that of the Brewin et al. (2015) model with its original model parameters for the global ocean (Fig. R7). It can be seen that the satellite-derived PSC from Chla using the Brewin et al. (2015) model were not consistent with the HPLC-derived values, showing clear overestimation for micro- phytoplankton and underestimation for pico-phytoplankton (as shown in dashed circles in Fig. R7b). The $R$ values of 0.57, 0.093, and 0.52 and RMSE values of 0.26, 0.15, and 0.21 were observed for micro-, nano-, and pico-phytoplankton, respectively. In contrast, the satellite-derived PSC using our refined PSC model agreed well with the HPLC-derived values, with the $R$ values of 0.68, 0.46, and 0.64 and RMSE values of 0.13, 0.13, and 0.19 for micro-, nano-, and pico- phytoplankton, respectively (Fig. R5a). These results indicated the performance of our refined PSC model using the reconstructed satellite data was better than that of the Brewin et al. (2015) model in our study region.

[Figure]

Fig. R7. Comparison of HPLC-derived and satellite-derived PSC data from the reconstructed satellite $R_{rs}$ using the refined PSC model in this study (a); from Chla using the Brewin et al. (2015) model (b).

Further, we have assessed and compared the performances of our refined PSC model and the Brewin et al. (2015) model for spring bloom water by showing the monthly climatologies of micro-phytoplankton size fraction ($f_{micro}$) during 2003-2016 in the month area of Changjiang river (MCJR) and middle shelf region (MSR), as shown in Fig. R8.

Regional averages of the monthly mean Chla in the MCJR (Fig. R8a) indicated two peaks. One peak of the mean Chla was found in summer (July), and it is likely to be related to the increase of nutrient from riverine discharge (He et al., 2013) (see detailed discussion in Section 4.2.1 in the revised manuscript). The other peak of the mean Chla occurred in spring (April-May). In the MCJR, the $f_{\text{micro}}$ derived using our PSC model were similar to those obtained from the Brewin et al. (2015) model in the winter and spring; however, the $f_{\text{micro}}$ of our model were lower than those of the Brewin et al. (2015) model in the summer and autumn. This may be related to the overestimation of micro-phytoplankton caused by the Brewin et al. (2015) model itself (see Fig. R7b). Furthermore, micro-phytoplankton of our refined PSC model comprised 60-80% of the Chla throughout the year with the maximum value in the spring (April), while $f_{\text{micro}}$ of the Brewin et al. (2015) model varied from 70% to 90% with the maximum value in the spring (May). In contrast to the MCJR, the MSR showed the lowest mean Chla in the summer, and the highest value in the spring (April) (Fig. R8b). The $f_{\text{micro}}$ temporal validations of our refined PSC model and the Brewin et al. (2015) model were similar to those of Chla, with the high $f_{\text{micro}}$ during spring and the low during summer and autumn.

Overall, both the two PSC models can capture the variation feature of PSC for spring bloom waters (Fig. R8). However, the retrieval accuracy of satellite-derived PSC using our refined PSC model was better than using the Brewin et al. (2015) model (Fig. R7). Therefore, in general, the results of Fig. R7 and Fig. R8 indicated that the performance of our refined PSC model for the PSC retrieval for spring bloom waters was better than that of the Brewin et al. (2015) model.

[Figure]

Fig. R8. Monthly climatology (2003-2016) of the mean MODIS-Aqua Chla (mg m$^{-3}$) and micro-phytoplankton size fraction in the MCJR (a) and MSR (b).

(5) Results shown in Fig.9 is very interesting, which have already attracted much attentions from marine

ecologists. I recommend authors to do further discussion about these variabilities referring those published results. At the same time, those published results about the spatial and seasonal variations of PSCs in ECS could be used for validating the MODIS derived values.

Response: Thank you for your comments. As you suggested, in the revised manuscript, we have added detailed discussion of the monthly climatological PSC in three subareas (shown in Fig. 9 in the revised manuscript). More details were provided in Section 4.2 in the revised manuscript (Section 4.2.1 - The coastal region; Section 4.2.2 - The middle shelf and shelf break of the ECS; Section 4.2.3 - The Kuroshio region and open ocean). Meanwhile, we have compared and discussed the spatiotemporal variations of phytoplankton size classes and community in the ECS reported from field measurements by other researchers (see Section 4.2 in the revised manuscript). The distribution patterns of satellite-derived PSC in the ECS in our study are generally consistent with those field PSC results, which suggested that the satellite-derived PSC in our study are generally reasonable. Furthermore, in the revised manuscript, we have added the explanation about the monthly climatological Chla and PSC in three subareas selected in this study (see the second paragraph of Section 3.6), and also added discussion about the related physical and biochemical effects for helping to understand the spatiotemporal variability of the PSC in the ECS (see Section 4.2). Thank you.

(6) Temperature itself is an important factor governing the distribution of phytoplankton, which also provides a quantitative index of the physio-chemical state of the marine environment. How about the correlation between total Chl-a (phytoplankton biomass) with SST? As shown in Table 5, these correlation between SST and size fractions may have different underlying mechanism for the 3 different subareas. Some more explanations about the hydrological backgrounds of these subareas are expected to deepen the understanding.

Response: Thank you for your comments. Based on the 14-years (2003-2016) time series of the monthly SST and Chla data, we have discussed the correlation between Chla with SST for the three subareas (Fig. R9). In the Kuroshio region, there was significant negative correlation ($R = -0.84$ and $p < 0.001$). The correlation was negative in the MSR ($R = -0.36$ and $p < 0.001$) , however it became positive in the MCJR ($R = 0.43$ and $p < 0.001$). These findings were in agreement with the study of Liu et al. (2013), which reported that there was a significantly positive correlation between Chla and SST at the sea region

with water depth < 20 m but a negative correlation at the sea region with water depth of 20 – 40 m and > 40 m.

[Figure]

Fig. R9 The scatterplots showing the relationship between the Chla and SST in the MCJR, MSR, and KR.

Meanwhile, in the revised manuscript, we have added the discussion of the hydrological backgrounds in the three subareas (Fig. R10; i.e., Fig. 10 in the revised manuscript), aiming to better understand the changes of PSC response to the SST variations. In the Kuroshio region, significant negative correlation between nano-phytoplankton size fraction and SST was found ($R$=-0.66 < -0.5 and $p$<0.001), and weak negative correlation was found for micro-phytoplankton ($R$=-0.31). Significant positive correlation between pico-phytoplankton size fraction and SST was identified ($R$=0.64 >0.5 and $p$<0.001) (Fig. R10c). Several studies have found that surface warming can weaken vertical mixing due to the increase in water column stability (Behrenfeld et al., 2006; Boyce et al., 2010), which causes less nutrient supply to the surface layers from underlying nutrient-rich waters. In addition, the Kuroshio water is characterized by high salinity, high temperature, and low nutrient (Jiao et al., 2005). These oligotrophic conditions favour the presence of smaller-sized phytoplankton (pico) and restrict the growth of larger-sized phytoplankton (micro and nano). It offers an explanation to help us for understanding the correlation between the increasing trend of SST and decreasing trend of micro- and nano-phytoplankton size fraction and increasing trend of pico-phytoplankton size fraction.

Similar to the KR, there were a negative correlation between micro-phytoplankton proportion and SST

($R$=-0.76) and a positive correlation between pico-phytoplankton proportion and SST ($R$=0.65) in the MSR (Fig. R10b). Different environmental conditions in the two subareas showed similar responses of the variability of micro- and pico-phytoplankton size fractions to SST. However, the increasing trend of SST and increasing trend of nano-phytoplankton showed a weak positive correlation ($R$=0.34) (Fig. R10b), which was different from the Kuroshio region. The weak correlation suggested that nano-phytoplankton in this region may be affected by factors (e.g., grazing-nitrogen rate) other than SST. For instance, Barlow et al. (2016) reported that nano-phytoplankton (e.g, flagellates) were dominant offshore in warmer shelf region, because they are better utilising the increase in nutrient concentrations after upwelled water has warmed.

In the mouth area of Changjiang River, the SST were negatively ($R$=-0.59), positively ($R$=0.58), and positively ($R$=0.54) correlated with micro-, nano-, and pico-phytoplankton size fractions respectively (Fig. R10a). The water body in coastal region mixes well in winter with low SST and has a weak stratification of water column in summer with high SST, as the hydrodynamic in coastal water is dominated by the variation of wind-tide-thermohaline circulations (Guan, 1994). In some degree, increasing of SST could result in the decrease of larger-sized phytoplankton (micro and nano) and the increase of smaller-sized phytoplankton (pico). However, the trend of rising SST and increasing nano-phytoplankton size fraction in the MCJR was observed. This may be related to the optimum temperature for the growth of different algal groups. Additionally, pervious studies have shown that the nutrient structure in the ECS have altered by Changjiang discharge, especially for the Changjiang estuary and adjacent area (Zhang et al., 2007; Wang et al., 2014). The change of nutrient structure (increase in N/P ratios) might be an important factor that affects the phytoplankton community in the MCJR. These results suggested the interannual variability of PSC in coastal waters is more complicated than in offshore waters. The detailed study focusing on the reasons of the changes of PCS in MCJR is still required.

Overall, the correlations between PSC and SST (Fig. R10) indicated SST is an important factor influencing the PSC dynamic in the ECS. The interannual variations of phytoplankton size classes in the ECS were complicated and could not be fully explained by the individual factor. In the revised manuscript, we have added more explanations regarding this issue in Section 4.3.

[Figure]

Fig. R10. The scatterplots showing the relationships between the monthly phytoplankton size fractions and SST from 2003 to 2016 for the MCJR (a), MSR (b), and KR (c).

Some specific technical suggestions:

5   1. Fig.1, Mean $R_{rs}$ spectra for coastal waters of ECS could be helpful for reader to know exactly the ocean color variability in ECS (which covers many water types).

Response: Thank you for your suggestions. In order to better show the ocean color variability in the ECS, we have analyzed the mean and coefficient of variation (CV) of $R_{rs}(\lambda)$ collected in the coastal region of Zhejiang (Zhe) and Fujian (Min) and southern Jeju Island (Fig. R11; i.e., Fig .2 in the revised

10  manuscript). The *in situ* $R_{rs}(\lambda)$ of all samples in the different georical locations exhibited large variability in both magnitudes and spectral shapes (Fig. R11 a and b). For the samples in the coastal region of Zhe-Min, both 10 wavebands showed larger variability in $R_{rs}(\lambda)$ magnitude with CV larger than 55% (Fig. R11 c). For the samples in southern Jeju Island, CV varied from 20% to 60%, with a minimum around 531 nm and 547 nm (Fig. R11 d). Overall, they showed large dynamic range with

15  significant variability. Additionally, we have added the description of this issue in the revised manuscript

(see the first paragraph of Section 2.1).

[Figure]

Fig. R11. $R_{rs}(\lambda)$ spectra at MODIS wavelengths collected in the coastal region of Zhe-Min (a) and southern Jeju Island (b); Mean spectra and coefficient of variation (CV) of $R_{rs}(\lambda)$ in the coastal region of Zhe-Min (c) and southern Jeju Island (d). The CV is derived as the standard deviation (SD) over the mean..

2. Coefficients of $K$ in Equation (9) could be shown in a Table.

Response: In the revised manuscript, the coefficients of $K$ in the Equation (9) and the corresponding statistical indicators ($R^2$, RMSE, and MAPE) were shown in Table 3. Meanwhile, Fig. 5 in the manuscript has been removed accordingly. Thank you.

3. Fig.8, add the spatial distribution of total Chla for 4 seasons.

Response: Thank you. In the revised manuscript, we have added the seasonal distributions of Chla in four seasons in Fig. 8 (Fig. R12 in this response). Additionally, we have also added the explanations regarding this issue in Section 3.5 in the revised manuscript.

[Figure]

Fig. R12. Seasonal distributions of the PSC (*a - l*) and Chla (A-D, right panel) in the ECS during 2003-2016.

4. Show locations for the three subareas in Fig.8 and introduce the box size in Data and Methods.

Response: Thank you for your valuable suggestion. In the revised manuscript, we have shown the locations for the three subareas in Fig. 1a and Fig. 8 (see Figs. R12 and R13 in this response). Additionally, as you suggested, we have added the description of these subareas (e.g., box size) in the Materials and Methods (see Section 2.1 in the revised manuscript).

[Figure]

Fig. R13. Distribution of *in situ* and matchup dataset and locations of the selected subareas (black boxes) (*a*), namely MCJR (mouth area of Changjiang river), MSR (middle shelf region), and KR (Kuroshio region); locations of sampling stations collected in the North Pacific and North Atlantic oceans from the NASA SeaBASS archive (b); the average satellite $R_{rs}(\lambda)$

spectral from 2003 to 2016 for the Kuroshio region, North Pacific ocean, and North Atlantic ocean (blue circles in b) (c). Error bars represent standard deviations of the means.

5. Fig.9, enlarge the y-axis of (c) and (d) for total Chl-a for clarity.

Response: To clearly show the y-axis for total Chl-a, this figure has been redrawn in the revised manuscript (see Fig. 9 in the revised manuscript, i.e., Fig. R14 in this response). Thank you.

[Figure]

Fig. R14. Monthly climatological PSC and Chla from 2003 to 2016 in the mouth area of Changjiang river (MCJR) (*a*), middle shelf region (MSR) (c), and the Kuroshio region (KR) (c). Error bars indicate standard deviations of the means.

6. For results shown in Table 5, a figure showing time series of size fractions and SST may be helpful for discussing their correlations.

Response: Thank you for your suggestion. In the revised manuscript, we have added the correlations between phytoplankton size fractions and SST for different subareas, and also their correlation coefficient *R* values and *p* values (see Fig. 10 in the revised manuscript, i.e., Fig. R15 in this response). In addition, Table 5 in the manuscript has been removed accordingly in the revised manuscript.

[Figure]

Fig. R15. The scatterplots showing the relationships between the monthly phytoplankton size fractions and SST from 2003 to 2016 for the MCJR (a), MSR (b), and KR (c).

**References used in this response:**

5    Barlow, R., Gibberd, M., Lamont, T., Aiken, J., and Holligan, P.: Chemotaxonomic phytoplankton patterns on the eastern boundary of the Atlantic Ocean, Deep Sea Research Part I: Oceanographic Research Papers, 111, 73-78, 2016.

Behrenfeld, M. J., O'Malley, R. T., Siegel, D. A., McClain, C. R., Sarmiento, J. L., Feldman, G. C., Milligan, A. J., Falkowski, P. G., Letelier, R. M., and Boss, E. S.: Climate-driven trends in contemporary ocean productivity,
10    Nature, 444, 752-755, 2006.

Boyce, D. G., Lewis, M. R., and Worm, B.: Global phytoplankton decline over the past century, Nature, 466, 591-596, 2010.

Brewin, R. J. W., Sathyendranath, S., Jackson, T., Barlow, R., Brotas, V., Airs, R., and Lamont, T.: Influence of light in the mixed-layer on the parameters of a three-component model of phytoplankton size class, Remote Sensing
15    of Environment, 168, 437-450, 10.1016/j.rse.2015.07.004, 2015.

Gregg, W. W., and Casey, N. W.: Global and regional evaluation of the SeaWiFS chlorophyll data set, Remote Sensing of Environment, 93, 463-479, 2004.

Guan, B.: Patterns and structures of the currents in Bohai, Huanghai and East China Seas, in: Oceanology of China seas, Springer, 17-26, 1994.

He, X., Bai, Y., Pan, D., Huang, N., Dong, X., Chen, J., Chen, C.-T. A., and Cui, Q.: Using geostationary satellite ocean color data to map the diurnal dynamics of suspended particulate matter in coastal waters, Remote Sensing of Environment, 133, 225-239, 2013.

Hu, C., Lee, Z., and Franz, B.: Chlorophyll aalgorithms for oligotrophic oceans: A novel approach based on three‐band reflectance difference, Journal of Geophysical Research: Oceans, 117, 2012.

Jiao, N., Yang, Y., Hong, N., Ma, Y., Harada, S., Koshikawa, H., and Watanabe, M.: Dynamics of autotrophic picoplankton and heterotrophic bacteria in the East China Sea, Continental Shelf Research, 25, 1265-1279, 2005.

Liu, D., and Wang, Y.: Trends of satellite derived chlorophyll-a (1997-2011) in the Bohai and Yellow Seas, China: Effects of bathymetry on seasonal and inter-annual patterns, Progress in Oceanography, 116, 154-166, 10.1016/j.pocean.2013.07.003, 2013.

Siswanto, E., Tang, J., Yamaguchi, H., Ahn, Y.-H., Ishizaka, J., Yoo, S., Kim, S.-W., Kiyomoto, Y., Yamada, K., and Chiang, C.: Empirical ocean-color algorithms to retrieve chlorophyll-a, total suspended matter, and colored dissolved organic matter absorption coefficient in the Yellow and East China Seas, Journal of oceanography, 67, 627, 2011.

Sun, D., Huan, Y., Qiu, Z., Hu, C., Wang, S., and He, Y.: Remote-Sensing Estimation of Phytoplankton Size Classes From GOCI Satellite Measurements in Bohai Sea and Yellow Sea, Journal of Geophysical Research: Oceans, 122, 8309-8325, 10.1002/2017jc013099, 2017.

Wang, S. Q., Ishizaka, J., Yamaguchi, H., Tripathy, S. C., Hayashi, M., Xu, Y. J., Mino, Y., Matsuno, T., Watanabe, Y., and Yoo, S. J.: Influence of the Changjiang River on the light absorption properties of phytoplankton from the East China Sea, Biogeosciences, 11, 1759-1773, 10.5194/bg-11-1759-2014, 2014.

Zhang, J., Liu, S., Ren, J., Wu, Y., and Zhang, G.: Nutrient gradients from the eutrophic Changjiang (Yangtze River) Estuary to the oligotrophic Kuroshio waters and re-evaluation of budgets for the East China Sea Shelf, Progress in Oceanography, 74, 449-478, 2007.

**Response to Reviewer #2**

The authors used *in situ* datasets and reconstructed $R_{rs}$ data from MODIS to estimate the PSC in the East China Sea and investigated the seasonal variability of the PSC in the ECS using ~14 years MODIS- $R_{rs}$ derived PSC data. The authors tuned the PCA approach proposed in an earlier study by Wang et al. 2014 to derive PSC from absorption measurements. The tuned approach was also applied to MODIS data via reconstructing MODIS $R_{rs}$ in the blue bands and the QAA inversion method for MODIS derived phytoplankton absorption. Improvements in methods led to better retrievals in this region, which is encouraging. Seasonality of PSC was also investigated by discussing the relating factors such as water vertical structure, temperature, upwelling, etc., providing a better understanding on the PSC in this region with regard to environmental changes. The manuscript was overall well written despite a few inconsistencies in the tense and wording. The authors have also improved the manuscript according to what I suggested in the first review, including selecting a few typical subregions and analyzing the climatological variation besides the seasonality.

Response: Thank you for your positive comment.

My further suggestions are as below and details were highlighted in the manuscript.

1 As phytoplankton absorption and Chla are quite important in PSC estimation, probably the authors can also display the seasonal distributions of QAA derived $a_{ph}$ (for example $a_{ph}(440)$ as $a_{ph}(675)$ is not successfully estimated) and QAA derived aph from reconstructed MODIS $R_{rs}$ in the ECS? As QAA only retrieves IOPs but not Chla, this can at least give a hint on how the Chla distributes and changes over seasons by showing $a_{ph}$ distributions. I then noticed that MODIS Chla products were also used. How is the MODIS Chla compared with the in situ ones at matchups? and how was the MODIS derived $a_{ph}$ versus in situ Chla? compared to the MODIS Chla, does QAA $a_{ph}(440)$ had a better correlation with *in situ* Chla?

Response: Thank you for your suggestions. First, we have discussed the relationships between $a_{ph}(443)$ and Chla for *in situ* measured and satellite datasets (Fig. R1). Here, the satellite $a_{ph}(443)$ were derived from the reconstructed satellite $R_{rs}$ data. From Fig. R1, the significant positive correlations ($R^2 > 0.9$ and $p < 0.001$) between the Chla and $a_{ph}(443)$ for both satellite and *in situ* measured dataset. Meanwhile, we

have analyzed the seasonal distributions of Chla and satellite-derived $a_{ph}(443)$ from the reconstructed satellite $R_{rs}$ data in the ECS for four season, as shown in Fig. R2. The seasonal distribution patterns of Chla in the ECS were generally similar to those of the satellite-derived $a_{ph}(443)$. Meanwhile, distribution of phytoplankton biomass may help us to explain the spatial variability of PSCs; therefore, we have added the seasonal pattern distributions of Chla in the revised manuscript (see Fig. 8 in the revised manuscript, i.e., Fig. R3 in this response).

Additionally, based on 22 satellite matchups, we have also assessed the accuracy of MODIS Chla data using *in situ* measured Chla data (Fig. R4). The satellite Chla data in the matchup dataset were obtained from the daily Level 2 Chla products from MODIS Aqua sensor. This match-up dataset only consisted of satellite Chla with an overpass time window within 5h before and after field data. To avoid the effects of outliers, the median Chla values for a 3×3 pixels window centered on the locations of the sampling stations were defined as satellite Chla. As shown in Fig. R4, satellite Chl-a data generally agreed well with *in situ* measured Chl-a, with the $R^2$, RMSE and MAPE values of 0.85, 0.16 mg·m$^{-3}$ and 31.38%, respectively. These results suggested that the MODIS Chl-a had high accuracy, which was considered generally acceptable in remote sensing research (Gregg and Casey, 2004; Siswanto et al., 2011).

Overall, in the revised manuscript, we have used the MODIS Chla products to help us to explain the spatial variability of PSC in the ECS (see Fig. 8 in the revised manuscript, i.e., Fig. R3 in this response). Meanwhile, in the revised manuscript, we have added the explanations regarding this issue in Section 3.5.

[Figure]

Fig. R1 Correlations between *in situ* measured Chla and the derived $a_{ph}(443)$ from *in situ* measured $R_{rs}$ (a); satellite Chla and satellite-derived $a_{ph}(443)$ from the reconstructed satellite $R_{rs}$ (b). Black lines correspond to the fit lines.

[Figure]

Fig. R2 Seasonal distributions of the satellite-derived $a_{ph}(443)$ from reconstructed $R_{rs}$ (a) and chlorophyll-a concentration (b) in the ECS during 2003-2016.

[Figure]

5    Fig. R3. Seasonal distributions of the PSC (*a-l*) and Chla (A-D, right panel) in the ECS during 2003-2016.

[Figure]

Fig. R4. Comparison of satellite-derived Chl-a with *in situ* measured values. Dashed line is the 1:1 line.

2 The authors stated that the findings presented here complement and enhance recent studies that have demonstrated that satellite ocean color data can be used to retrieve the PSC in the ECS. What other studies in this region or in China's seas? How are your results compared to these studies? are they basically in consistence with the others?

Response: Thank you. To our knowledge, there is no study to examine PSC distribution in the ECS at synoptic scales from satellite observations. Previous investigations on the PSC in the ECS have been conducted based on field observations (e.g., Chen, 2000; Furuya et al., 2003; Wang et al., 2015). However, there are some studies that have estimated the PSC distribution in other China seas (e.g., Bohai sea , Yellow sea, and South China Sea ) using the satellite ocean colour data. For instance, Sun et al. (2017) developed a local model to estimate PSC distributions in the Bohai Sea and Yellow Sea.

Meanwhile, we have compared and discussed satellite-derived PSC with previous field investigations in the ECS, and found that the distributions of PSC in our study were generally consistent with those reported by other researchers from field observations (Chen, 2000; Furuya et al., 2003; Wang et al., 2015). In addition, Fig. R5 further showed the comparison between the spatial distributions of PSC during summer and autumn derived from both MODIS and our field measurements. Overall, their general distributions patterns agreed with each other. For micro-phytoplankton, high values were generally found in near-shore regions with lower values in offshore waters during summer and autumn.

For nano-phytoplankton, both MODIS and field measurements showed high values in the middle shelf region of the ECS. For pico-phytoplankton, both MODIS and field measurements showed low values in the coastal region during summer and autumn and high values in the coastal waters of western Japan during summer. Overall, these results suggested that the refined PSC model in our study was able to derive reasonable PSC patterns in our study region.

[Figure]

Fig. R5. Comparison of spatial distributions of PSC in the ECS between satellite retrievals and field measurements during summer and autumn.

3 Description on statistics sometimes is not very precise. Such as the authors used 'acceptable errors' or 'significant correlation' but did not explain how you defined acceptable or significant, maybe P values help a bit or change the way of interpretation.

Response: Thank you for your suggestion. To precisely describe the statistics, we have revised the manuscript carefully based on your comments. For instance, we have rephrased the "acceptable errors" (see the last paragraph in Section 3.4). In addition, we have added $p$ values of the correlation in Fig. 10 in the revised manuscript (Fig. R6 in this response).

[Figure]

Fig. R6. The scatterplots showing the relationships between the monthly phytoplankton size fractions and SST from 2003 to 2016 for the MCJR (a), MSR (b), and KR (c).

4 The description of the sub-regions is inadequate, please specify their sizes, and use boxes to specify the exact size and location in the map.

Response: Thank you for your suggestion. In the revised manuscript, we have displayed the locations for the three subareas selected in our study in Fig. 1a (Fig. R7 in this response). Additionally, we have added the description of these subareas (e.g., box size) in the Materials and Methods (see Section 2.1 in the revised manuscript).

[Figure]

Fig. R7. Distribution of *in situ* and matchup dataset and locations of the selected subareas (black boxes) (*a*), namely MCJR (mouth area of Changjiang river), MSR (middle shelf region), and KR (Kuroshio region); locations of sampling stations collected in the North Pacific and North Atlantic oceans from the NASA SeaBASS archive (b); the average satellite $R_{rs}(\lambda)$ spectral from 2003 to 2016 for the Kuroshio region, North Pacific ocean, and North Atlantic ocean (blue circles in b) (c). Error bars represent standard deviations of the means.

5 When discussing the PSC response to the SST, it might be also helpful to also show the SST variations.

Response: In the revised manuscript, we have added the correlations between phytoplankton size fractions and SST for different subareas (see Fig. 10 in the revised manuscript, i.e., Fig. R6 in this response). In addition, Table 5 in the manuscript has been removed accordingly. Thank you very much.

More detailed comments and technical corrections were listed in the attached file. Please also note the supplement to this comment:

https://www.biogeosciences-discuss.net/bg-2017-508/bg-2017-508-RC2- supplement.pdf

Response: Thank you for your valuable comments and suggestions. We have carefully revised all of these issues according to your detailed comments in the PDF file.

**References used in this response:**

Chen, Y. L. L.: Comparisons of primary productivity and phytoplankton size structure in the marginal regions of southern East China Sea, Continental Shelf Research, 20, 437-458, 2000.

Furuya, K., Hayashi, M., Yabushita, Y., and Ishikawa, A.: Phytoplankton dynamics in the East China Sea in spring and summer as revealed by HPLC-derived pigment signatures, Deep Sea Research Part II: Topical Studies in Oceanography, 50, 367-387, 2003.

Gregg, W. W., and Casey, N. W.: Global and regional evaluation of the SeaWiFS chlorophyll data set, Remote Sensing of Environment, 93, 463-479, 2004.

Siswanto, E., Tang, J., Yamaguchi, H., Ahn, Y.-H., Ishizaka, J., Yoo, S., Kim, S.-W., Kiyomoto, Y., Yamada, K., and Chiang, C.: Empirical ocean-color algorithms to retrieve chlorophyll-a, total suspended matter, and colored dissolved organic matter absorption coefficient in the Yellow and East China Seas, Journal of oceanography, 67, 627, 2011.

Sun, D., Huan, Y., Qiu, Z., Hu, C., Wang, S., and He, Y.: Remote‐Sensing Estimation of Phytoplankton Size Classes From GOCI Satellite Measurements in Bohai Sea and Yellow Sea, Journal of Geophysical Research: Oceans, 122, 8309-8325, 2017.

Wang, S., Ishizaka, J., Hirawake, T., Watanabe, Y., Zhu, Y., Hayashi, M., and Yoo, S.: Remote estimation of phytoplankton size fractions using the spectral shape of light absorption, Opt Express, 23, 10301-10318, 10.1364/OE.23.010301, 2015.

---

## Referee Report (RR1)

Referee comment for bg-2017-508

The spectral-based model proposed by Wang et al. (2014) was modified in this study for retrieving PSC in the East China Sea from MODIS data. Internal relationship between PSC and the spectral variability of phytoplankton absorption is the key point for this model. In order to minimize the effect of high noise and low accuracy at shore wavelengths, the authors reconstructed the Rrs spectra by using the multivariate linear relationships at different wavelengths. Based on in situ match-up data analyses, satellite derived PSC compared well with those derived from HPLC pigment composition. Seasonal variability of PSC in the three sub-regions were discussed by considering different environmental factors, which gave us a better understanding of PSC distribution in ECS at synoptic scales. We can see obvious improvement in the revised manuscript.

I suggest minor revision considering the following points:

1. Detailed information for estimating PSC from the DPA approach are required. In this study, Chlorophyll-b was one of the diagnostic pigments of nano-phytoplankton, which is different from that method used by Brewin et al. (2010) for open ocean. More explanations are needed.

2. It's not surprising to see the poor performance of the model by Brewin et al. (2015) or Sun et al. (2017). There are several points we have to consider. Different criteria for estimating the PSC from the diagnostic pigments were used which may result in large differences in the basic dataset. We also have to consider the regional differences. Did the authors use the model directly? Maybe the coefficients for these models could be locally modified before comparison.

3. General spatial distribution of PSCs seems reasonable. Obvious differences about the seasonal variability of PSCs in three sub-regions were shown in Fig.8 and Fig.9. We can see clear shift of the dominant phytoplankton size class, especially in MSR and KR regions. More explanations about these variabilities with referred to previous work (two of them are listed below) could be very helpful for confirming these results. I think these results could also be highlighted in the abstract.

4. The exact size and location of three sub-regions could be specified by giving the longitude and latitude range, rather than the pixel numbers.

Some references:

Guo, S.J., Y.Y. Feng, L. Wang, M.H. Dai, Z.L. Liu, Y. Bai, and J. Sun, 2014. Seasonal variation in the phytoplankton community of a continental-shelf sea: the East China Sea. Marine Ecology Progress Series, 516 103-126

Liu, X., Xiao, W., Landry, M.R. et al. , 2016. Responses of Phytoplankton Communities to Environmental Variability in the East China Sea. Ecosystems , 19: 832-849. https://doi.org/10.1007/s10021-016-9970-5.

---

## Author Response (AR2)

Dear Reviewers,

We deeply appreciate your comments and effort towards improving our manuscript. We have taken your constructive comments carefully in the revision of our manuscript. For the revision, please kindly refer to the point-to-point responses as followings and the revised manuscript. The changes we made have been noted in the blue color for highlighting.

**Response to Reviewer**

The spectral-based model proposed by Wang et al. (2014) was modified in this study for retrieving PSC in the East China Sea from MODIS data. Internal relationship between PSC and the spectral variability of phytoplankton absorption is the key point for this model. In order to minimize the effect of high noise and low accuracy at shore wavelengths, the authors reconstructed the $R_{rs}$ spectra by using the multivariate linear relationships at different wavelengths. Based on in situ match-up data analyses, satellite derived PSC compared well with those derived from HPLC pigment composition. Seasonal variability of PSC in the three sub-regions were discussed by considering different environmental factors, which gave us a better understanding of PSC distribution in ECS at synoptic scales. We can see obvious improvement in the revised manuscript.

Response: Thank you for your positive comment.

I suggest minor revision considering the following points:

1. Detailed information for estimating PSC from the DPA approach are required. In this study, Chlorophyll-b was one of the diagnostic pigments of nano-phytoplankton, which is different from that method used by Brewin et al. (2010) for open ocean. More explanations are needed.

Response: Thank you for your suggestion. In our study, the diagnostic pigment analysis (DPA) was applied to compute the PSC from HPLC pigment data (hereafter called the HPLC-derived PSC). The DPA approach was originally proposed by Vidussi et al. (2001), and subsequently improved by Uitz et al. (2006). In addition, Hirata et al. (2008) used the improved DPA approach to account for the occurrence of Chlorophyll-b in nano-phytoplankton class size, because it was most abundant at high

Chla ($> 0.25$ mg m$^{-3}$) and was a minor pigment at lower Chla (Hirata et al., 2008). Subsequently, Hirata et al. (2011) and Brewin et al. (2010) further refined the DPA approach to account for ambiguity of $C_f$ signal in diatoms and the occurrence of $C_h$ signal in picophytoplankton. In this study, the HPLC-derived PSC results were then given by:

$$f_{micro} = \left(1.41C_f + 1.41C_p\right)/\sum W_i P_i \tag{1}$$

$$f_{nano} = \left(0.60C_a + 0.35C_b + 1.01C_{Cb} + x\times1.27C_h\right)/\sum W_i P_i \tag{2}$$

$$f_{pico} = \left(0.86C_z + y\times1.27C_h\right)/\sum W_i P_i \tag{3}$$

where $x$ and $y$ are the proportions of nano- and pico-phytoplankton in Hex, respectively. When Chla $>0.08$ mg m$^{-3}$, $x=1$ and $y=0$; when Chla are between 0.001 and 0.08 mg m$^{-3}$, $x=12.5$Chla and $y=1-12.5$Chla. $\sum W_i P_i$ is the weighted sum of the seven diagnostic pigments (Uitz et al., 2006), according to the formula:

$$\sum W_i P_i = 1.41C_f + 1.41C_p + 0.60C_a + 0.35C_b + 1.27C_h + 0.86C_z + 1.01C_{Cb} \tag{4}$$

In the revised manuscript, we have added these explanations in Section 2.2.1.

2. It's not surprising to see the poor performance of the model by Brewin et al. (2015) or Sun et al. (2017). There are several points we have to consider. Different criteria for estimating the PSC from the diagnostic pigments were used which may result in large differences in the basic dataset. We also have to consider the regional differences. Did the authors use the model directly? Maybe the coefficients for these models could be locally modified before comparison.

Response: Thank you for your valuable comments and suggestions. Indeed, we directly used the Brewin et al. (2015) and Sun et al. (2017) models with their original coefficients in the manuscript. The regional differences may affect and result in the poor performance of these models. Thus, in the revised manuscript, in order to better assess the performance of these models in the ECS, as you suggested, we regionally tuned these published models before comparison. The fitting produce was applied to our field dataset collected in the ECS using a standard nonlinear least-squares method. Fig. R1 and Fig. R2 show the comparison between the HPLC-derived PSC and modeled PSC values using the retuned Brewin et al. (2015) model and the retuned Sun et al. (2017) model, respectively. However, it can be clearly seen from Fig. R1 (top row) that the data were quite scattered, and the fractional variation of each population

in the ECS as a function of Chla didn't strictly agree with the results of Brewin et al. (2015). Meanwhile, the scatter plots of Fig. R1 (bottom row) and Fig. R2 show the poor fitting results of the regionally tuned model development with low $R$ values and high RMSE values, especially for nano-phytoplankton. These results indicated that the two "abundance-based" models may not perform well in the East China Sea as suggested by Wang et al. (2014). Therefore, it should be noted here that the two retuned models were used to better assess the performance of our refined PSC model only, although the Brewin et al. (2015) model and Sun et al. (2017) model may not necessarily applicable in the ECS (Wang et al., 2014).

[Figure]

Fig. R1 The top row shows the size fractions of phytoplankton ($f_i$) as a function of chlorophyll-a concentration, and the bottom row shows the comparison results between the HPLC-derived PSC and modeled PSC values using the retuned Brewin et al. (2015) model.

[Figure]

Fig. R2 The comparison results between the HPLC-derived PSC and modeled PSC values using the retuned Sun et al. (2017) model.

In this study, the retuned Brewin et al. (2015) model for the ECS was expressed as:

$$f_{pico} = 0.19\left[1 - exp\left(-3.6Chla\right)\right]/Chla$$
$$f_{p,n} = 1.0\left[1 - exp\left(-1.0Chla\right)\right]/Chla$$
$$f_{nano} = f_{p,n} - f_{pico} \tag{1}$$
$$f_{micro} = 1 - f_{p,n}$$

where $f_{micro}$, $f_{nano}$, $f_{pico}$, and $f_{p,n}$ are the size fractions of micro-, nano-, pico-phytoplankton, and the sum of nano- and pico-phytoplankton, respectively. Chla is the chlorophyll a concentration. And, the retuned Sun et al. (2017) model for the ECS was expressed as:

$$f_{pico} = 0.66Chla^{-1}\left[1 - exp\left(-Chla^2 \times R_{rs}\left(680\right)\right)\right]^{0.16}$$
$$f_{nano} = 4.17Chla^{-1}\left[1 - exp\left(-Chla^2 \times R_{rs}\left(680\right)\right)\right]^{0.32} \tag{2}$$
$$f_{micro} = 1 - f_{nano} - f_{pico}$$

where $R_{rs}$ (680) is the remote sensing reflectance at 680 nm.

Our refined PSC model was compared with the retuned Brewin et al. (2015) model and the retuned Sun et al. (2017) model (Fig. R3, i.e., Fig. 7 in the revised manuscript). The scatter distributions of the satellite-derived PSC using our refined PSC model were closer to the 1:1 line than those of the other two models. According to the statistical indicators, our refined PSC model had the best performance, with higher $R$ values and lower RMSE values (Fig. R3b). For the retuned Brewin et al. (2015) model, the $R$ values of 0.58, 0.066, and 0.53 and RMSE values of 0.2, 0.14, and 0.18 were observed for micro-, nano-, and pico-phytoplankton, respectively (Fig. R3c). For the retuned Sun et al. (2017) model, the $R$ values were 0.36, -0.042, 0.5 for micro-, nano-, and pico-phytoplankton, when the corresponding RMSE values were 0.25, 0.17, and 0.18, respectively (Fig. R3d). The retuned Brewin et al. (2015) model and the retuned Sun et al. (2017) model had relatively poor performance in the ECS. These comparison results indicated that the performance of our refined PSC model using the reconstructed satellite data was better than those of the retuned Brewin et al. (2015) model and the retuned Sun et al. (2017) model in our study region.

In the revised manuscript, we have added these explanations about this issue in Section 3.4.

[Figure]

Fig. R3. Comparison of the HPLC-derived and satellite-derived PSC values from the original satellite $R_{rs}$ (a) and the reconstructed satellite $R_{rs}^{rc}$ (b) using the refined PSC model in this study; using the retuned Brewin et al. (2015) model (c); using the retuned Sun et al. (2017) model (d). Solid lines denote the 1:1 lines.

5    3. General spatial distribution of PSCs seems reasonable. Obvious differences about the seasonal variability of PSCs in three sub-regions were shown in Fig.8 and Fig.9. We can see clear shift of the dominant phytoplankton size class, especially in MSR and KR regions. More explanations about these

variabilities with referred to previous work (two of them are listed below) could be very helpful for confirming these results. I think these results could also be highlighted in the abstract.

Some references:

Guo, S.J., Y.Y. Feng, L. Wang, M.H. Dai, Z.L. Liu, Y. Bai, and J. Sun, 2014. Seasonal variation in the phytoplankton community of a continental-shelf sea: the East China Sea. Marine Ecology Progress Series, 516 103-126. Liu, X., Xiao, W., Landry, M.R. et al.,2016. Responses of Phytoplankton Communities to Environmental Variability in the East China Sea. Ecosystems,19: 832-849. https://doi.org/10.1007/s10021-016-9970-5.

Response: Thank you for your comments and the valuable references. In the revised manuscript, we have added more detailed explanations and discussion of the seasonal distributions of the PSC in the ECS (see Section 4.2), and also added the related previous studies to support the findings of our study. Meanwhile, these references have been cited in our work accordingly. Furthermore, in the revised manuscript, the main results about the spatiotemporal variations of the PSC in the ECS have been highlighted in the Abstract. Thank you.

4. The exact size and location of three sub-regions could be specified by giving the longitude and latitude range, rather than the pixel numbers.

Response: Thank you for your suggestion. In the revised manuscript, we have given the longitude and latitude range of three subareas (see the last paragraph of Section 2.1).

**References used in this response:**

Brewin, R. J., Sathyendranath, S., Hirata, T., Lavender, S. J., Barciela, R. M., and Hardman-Mountford, N. J.: A three-component model of phytoplankton size class for the Atlantic Ocean, Ecological Modelling, 221, 1472-1483, 2010.

Brewin, R. J. W., Sathyendranath, S., Jackson, T., Barlow, R., Brotas, V., Airs, R., and Lamont, T.: Influence of light in the mixed-layer on the parameters of a three-component model of phytoplankton size class, Remote Sensing of Environment, 168, 437-450, 10.1016/j.rse.2015.07.004, 2015.

Hirata, T., Aiken, J., Hardman-Mountford, N., Smyth, T. J., and Barlow, R. G.: An absorption model to determine phytoplankton size classes from satellite ocean colour, Remote Sensing of Environment, 112, 3153-3159, 10.1016/j.rse.2008.03.011, 2008.

Hirata, T., Hardman-Mountford, N., Brewin, R., Aiken, J., Barlow, R., Suzuki, K., Isada, T., Howell, E., Hashioka,

T., and Noguchi-Aita, M.: Synoptic relationships between surface Chlorophyll-a and diagnostic pigments specific to phytoplankton functional types, Biogeosciences, 8, 311-327, 2011.

Sun, D., Huan, Y., Qiu, Z., Hu, C., Wang, S., and He, Y.: Remote‐Sensing Estimation of Phytoplankton Size Classes From GOCI Satellite Measurements in Bohai Sea and Yellow Sea, Journal of Geophysical Research: Oceans, 122, 8309-8325, 2017.

Uitz, J., Claustre, H., Morel, A., and Hooker, S. B.: Vertical distribution of phytoplankton communities in open ocean: An assessment based on surface chlorophyll, Journal of Geophysical Research: Oceans, 111, 2006.

Vidussi, F., Claustre, H., Manca, B. B., Luchetta, A., and Marty, J. C.: Phytoplankton pigment distribution in relation to upper thermocline circulation in the eastern Mediterranean Sea during winter, Journal of Geophysical Research: Oceans, 106, 19939-19956, 2001.

Wang, S. Q., Ishizaka, J., Yamaguchi, H., Tripathy, S. C., Hayashi, M., Xu, Y. J., Mino, Y., Matsuno, T., Watanabe, Y., and Yoo, S. J.: Influence of the Changjiang River on the light absorption properties of phytoplankton from the East China Sea, Biogeosciences, 11, 1759-1773, 2014.